# META-LEARNING ADAPTIVE DEEP KERNEL GAUSSIAN PROCESSES FOR MOLECULAR PROPERTY PREDICTION

**Wenlin Chen**
University of Cambridge
MPI for Intelligent Systems
wc337@cam.ac.uk

**Austin Tripp**
University of Cambridge
ajt212@cam.ac.uk

**José Miguel Hernández-Lobato**
University of Cambridge
jmh233@cam.ac.uk

## ABSTRACT

We propose Adaptive Deep Kernel Fitting with Implicit Function Theorem (ADKF-IFT), a novel framework for learning deep kernel Gaussian processes (GPs) by interpolating between meta-learning and conventional deep kernel learning. Our approach employs a bilevel optimization objective where we meta-learn generally useful feature representations across tasks, in the sense that task-specific GP models estimated on top of such features achieve the lowest possible predictive loss on average. We solve the resulting nested optimization problem using the implicit function theorem (IFT). We show that our ADKF-IFT framework contains previously proposed Deep Kernel Learning (DKL) and Deep Kernel Transfer (DKT) as special cases. Although ADKF-IFT is a completely general method, we argue that it is especially well-suited for drug discovery problems and demonstrate that it significantly outperforms previous state-of-the-art methods on a variety of real-world few-shot molecular property prediction tasks and out-of-domain molecular property prediction and optimization tasks.

## 1 INTRODUCTION

Many real-world applications require machine learning algorithms to make robust predictions with well-calibrated uncertainty given very limited training data. One important example is drug discovery, where practitioners not only want models to accurately predict biochemical/physicochemical properties of molecules, but also want to use models to guide the search for novel molecules with desirable properties, leveraging techniques such as Bayesian optimization (BO) which heavily rely on accurate uncertainty estimates (Frazier, 2018). Despite the meteoric rise of neural networks over the past decade, their notoriously overconfident and unreliable uncertainty estimates (Szegedy et al., 2013) make them generally ineffective surrogate models for BO. Instead, most contemporary BO implementations use Gaussian processes (GPs) (Rasmussen & Williams, 2006) as surrogate models due to their analytically-tractable and generally reliable uncertainty estimates, even on small datasets.

Traditionally, GPs are fit on hand-engineered features (e.g., molecular fingerprints), which can limit their predictive performance on complex, structured, high-dimensional data where designing informative features is challenging (e.g., molecules). Naturally, a number of works have proposed to improve performance by instead fitting GPs on features learned by a deep neural network: a family of models generally called Deep Kernel GPs. However, there is no clear consensus about how to train these models: maximizing the GP marginal likelihood (Hinton & Salakhutdinov, 2007; Wilson et al., 2016b) has been shown to overfit on small datasets (Ober et al., 2021), while meta-learning (Patacchiola et al., 2020) and fully-Bayesian approaches (Ober et al., 2021) avoid this at the cost of making strong, often unrealistic assumptions. This suggests that there is demand for new, better techniques for training deep kernel GPs.

In this work, we present a novel, general framework called *Adaptive Deep Kernel Fitting with Implicit Function Theorem* (ADKF-IFT) for training deep kernel GPs which we believe is especially well-suited to small datasets. ADKF-IFT essentially trains a subset of the model parameters with a meta-learning loss, and separately adapts the remaining parameters on each task using maximum marginal likelihood. In contrast to previous methods which use a single loss for all parameters, ADKF-IFT is able to utilize the implicit regularization of meta-learning to prevent overfitting while avoiding the strong assumptions of a pure meta-learning approach which may lead to underfitting. The key contributions and outline of the paper are as follows:

1. As our main technical contribution, we present the general ADKF-IFT framework and its natural formulation as a bilevel optimization problem (Section 3.1), then explain how the implicit function theorem (IFT) can be used to efficiently solve it with gradient-based methods in a few-shot learning setting (Section 3.2).

2. We show how ADKF-IFT can be viewed as a *generalization* and *unification* of previous approaches based purely on single-task learning (Wilson et al., 2016b) or purely on meta-learning (Patacchiola et al., 2020) for training deep kernel GPs (Section 3.3).

3. We propose a specific practical instantiation of ADKF-IFT wherein all feature extractor parameters are meta-learned, which has a clear interpretation and obviates the need for any Hessian approximations. We argue why this particular instantiation is well-suited to retain the best properties of previously proposed methods (Section 3.4).

4. Motivated by the general demand for better GP models in chemistry, we perform an extensive empirical evaluation of ADKF-IFT on several chemical tasks, finding that it significantly improves upon previous state-of-the-art methods (Section 5).

## 2 BACKGROUND AND NOTATION

**Gaussian Processes** (GPs) are tools for specifying Bayesian priors over functions (Rasmussen & Williams, 2006). A $\mathcal{GP}(m_{\boldsymbol{\theta}}(\cdot), c_{\boldsymbol{\theta}}(\cdot, \cdot))$ is fully specified by a mean function $m_{\boldsymbol{\theta}}(\cdot)$ and a symmetric positive-definite covariance function $c_{\boldsymbol{\theta}}(\cdot, \cdot)$. The covariance function encodes the inductive bias (e.g., smoothness) of a GP. One advantage of GPs is that it is easy to perform principled model selection for its hyperparameters $\boldsymbol{\theta} \in \Theta$ using the marginal likelihood $p(\mathbf{y} \mid \mathbf{X}, \boldsymbol{\theta})$ evaluated on the training data $(\mathbf{X}, \mathbf{y})$ and to obtain closed-form probabilistic predictions $p(\mathbf{y}_* \mid \mathbf{X}_*, \mathbf{X}, \mathbf{y}, \boldsymbol{\theta})$ for the test data $(\mathbf{X}_*, \mathbf{y}_*)$; we refer the readers to Rasmussen & Williams (2006) for more details.

**Deep Kernel Gaussian Processes** are GPs whose covariance function is constructed by first using a neural network *feature extractor* $\mathbf{f}_{\boldsymbol{\phi}}$ with parameters $\boldsymbol{\phi} \in \Phi$ to create feature representations $\mathbf{h} = \mathbf{f}_{\boldsymbol{\phi}}(\mathbf{x}), \mathbf{h}' = \mathbf{f}_{\boldsymbol{\phi}}(\mathbf{x}')$ of the input points $\mathbf{x}, \mathbf{x}'$, then feeding these feature representations into a standard *base kernel* $c_{\boldsymbol{\theta}}(\mathbf{h}, \mathbf{h}')$ (e.g., an RBF kernel) (Hinton & Salakhutdinov, 2007; Wilson et al., 2016b;a; Bradshaw et al., 2017; Calandra et al., 2016). The complete covariance function is therefore $k_{\boldsymbol{\psi}}(\mathbf{x}, \mathbf{x}') = c_{\boldsymbol{\theta}}(\mathbf{f}_{\boldsymbol{\phi}}(\mathbf{x}), \mathbf{f}_{\boldsymbol{\phi}}(\mathbf{x}'))$ with learnable parameters $\boldsymbol{\psi} = (\boldsymbol{\theta}, \boldsymbol{\phi})$.

**Few-shot Learning** refers to learning on many related tasks when each task has few labelled examples (Miller et al., 2000; Lake et al., 2011). In the standard problem setup, one is given a set of training tasks $\mathcal{D} = \{\mathcal{T}_t\}_{t=1}^{T}$ (a *meta-dataset*) and some unseen test tasks $\mathcal{D}_* = \{\mathcal{T}_*\}$. Each task $\mathcal{T} = \{(\mathbf{x}_i, y_i)\}_{i=1}^{N_{\mathcal{T}}}$ is a set of points in the domain $\mathcal{X}$ (e.g., space of molecules) with corresponding labels (continuous, categorical, etc.), and is partitioned into a *support set* $\mathcal{S}_{\mathcal{T}} \subseteq \mathcal{T}$ for training and a *query set* $\mathcal{Q}_{\mathcal{T}} = \mathcal{T} \setminus \mathcal{S}_{\mathcal{T}}$ for testing. Typically, the total number of training tasks $T = |\mathcal{D}|$ is large, while the size of each support set $|\mathcal{S}_{\mathcal{T}}|$ is small. Models for few-shot learning are typically trained to accurately predict $\mathcal{Q}_{\mathcal{T}}$ given $\mathcal{S}_{\mathcal{T}}$ for $\mathcal{T} \in \mathcal{D}$ during a *meta-training* phase, then evaluated by their prediction error on $\mathcal{Q}_{\mathcal{T}_*}$ given $\mathcal{S}_{\mathcal{T}_*}$ for unseen test tasks $\mathcal{T}_* \in \mathcal{D}_*$ during a *meta-testing* phase.

## 3 ADAPTIVE DEEP KERNEL FITTING WITH IMPLICIT FUNCTION THEOREM

### 3.1 THE GENERAL ADKF-IFT FRAMEWORK FOR LEARNING DEEP KERNEL GPS

Let $A_{\Theta}$ and $A_{\Phi}$ respectively be the sets of base kernel and feature extractor parameters for a deep kernel GP. Denote the set of all parameters by $A_{\Psi} = A_{\Theta} \cup A_{\Phi}$. The key idea of the general ADKF-IFT framework is that only a subset of the parameters $A_{\Psi_{\text{adapt}}} \subseteq A_{\Psi}$ will be adapted to each individual task by minimizing a train loss $\mathcal{L}_T$, with the remaining set of parameters $A_{\Psi_{\text{meta}}} = A_{\Psi} \setminus A_{\Psi_{\text{adapt}}}$ meta-learned during a *meta-training* phase to yield the best possible validation loss $\mathcal{L}_V$ *on average* over many related training tasks (*after* $A_{\Psi_{\text{adapt}}}$ is separately adapted to each of these tasks). This can be naturally formalized as the following *bilevel optimization* problem:

$$\boldsymbol{\psi}_{\text{meta}}^* = \underset{\boldsymbol{\psi}_{\text{meta}}}{\arg\min} \ \mathbb{E}_{p(\mathcal{T})}[\mathcal{L}_V(\boldsymbol{\psi}_{\text{meta}}, \boldsymbol{\psi}_{\text{adapt}}^*(\boldsymbol{\psi}_{\text{meta}}, \mathcal{S}_{\mathcal{T}}), \mathcal{T})], \quad (1)$$

$$\text{such that} \quad \boldsymbol{\psi}_{\text{adapt}}^*(\boldsymbol{\psi}_{\text{meta}}, \mathcal{S}_{\mathcal{T}}) = \underset{\boldsymbol{\psi}_{\text{adapt}}}{\arg\min} \ \mathcal{L}_T(\boldsymbol{\psi}_{\text{meta}}, \boldsymbol{\psi}_{\text{adapt}}, \mathcal{S}_{\mathcal{T}}). \quad (2)$$

Equations (1) and (2) are most easily understood by separately considering the meta-learned parameters $\boldsymbol{\psi}_{\text{meta}}$ and the task-specific parameters $\boldsymbol{\psi}_{\text{adapt}}$. For a given task $\mathcal{T}$ and an arbitrary value for the meta-learned parameters $\boldsymbol{\psi}_{\text{meta}}$, in Equation (2) the task-specific parameters $\boldsymbol{\psi}_{\text{adapt}}$ are chosen to minimize the *train loss* $\mathcal{L}_T$ evaluated on the task's support set $\mathcal{S}_{\mathcal{T}}$. That is, $\boldsymbol{\psi}_{\text{adapt}}$ is *adapted* to the support set $\mathcal{S}_{\mathcal{T}}$ of the task $\mathcal{T}$, with the aim of producing the best possible model on $\mathcal{S}_{\mathcal{T}}$ for the given value of $\boldsymbol{\psi}_{\text{meta}}$. The result is a model with optimal task-specific parameters $\boldsymbol{\psi}^*_{\text{adapt}}(\boldsymbol{\psi}_{\text{meta}}, \mathcal{S}_{\mathcal{T}})$ for the given meta-learned parameters $\boldsymbol{\psi}_{\text{meta}}$ and task $\mathcal{T}$. The remaining question is how to choose a value for the meta-learned parameters $\boldsymbol{\psi}_{\text{meta}}$, knowing that $\boldsymbol{\psi}_{\text{adapt}}$ will be adapted separately to each task. In Equation (1), we propose to choose $\boldsymbol{\psi}_{\text{meta}}$ to minimize the expected *validation loss* $\mathcal{L}_V$ over a distribution of training tasks $p(\mathcal{T})$. There are two reasons for this. First, on any given task $\mathcal{T}$, the validation loss usually reflects the performance metric of interest on the query set $\mathcal{Q}_{\mathcal{T}}$ of $\mathcal{T}$ (e.g., the prediction error). Second, because the same value of $\boldsymbol{\psi}_{\text{meta}}$ will be used for all tasks, it makes sense to choose a value whose *expected performance* is good across many tasks drawn from $p(\mathcal{T})$. That is, $\boldsymbol{\psi}_{\text{meta}}$ is chosen such that a GP achieves the lowest possible average validation loss on the query set $\mathcal{Q}_{\mathcal{T}}$ of a random training task $\mathcal{T} \sim p(\mathcal{T})$ after $\boldsymbol{\psi}_{\text{adapt}}$ is adapted to the task's support set $\mathcal{S}_{\mathcal{T}}$.

In practice, $\boldsymbol{\psi}_{\text{meta}}$ would be optimized during a *meta-training* phase using a set of training tasks $\mathcal{D}$ to approximate Equation (1). After meta-training (i.e., at meta-test time), we make predictions for each unseen test task $\mathcal{T}_*$ using the joint GP posterior predictive distribution with optimal parameters $\boldsymbol{\psi}^*_{\text{meta}}$ and $\boldsymbol{\psi}^*_{\text{adapt}}(\boldsymbol{\psi}^*_{\text{meta}}, \mathcal{S}_{\mathcal{T}_*})$:

$$p(\mathcal{Q}^y_{\mathcal{T}_*} \mid \mathcal{Q}^{\mathbf{x}}_{\mathcal{T}_*}, \mathcal{S}_{\mathcal{T}_*}, \boldsymbol{\psi}^*_{\text{meta}}, \boldsymbol{\psi}^*_{\text{adapt}}(\boldsymbol{\psi}^*_{\text{meta}}, \mathcal{S}_{\mathcal{T}_*})). \tag{3}$$

Note that the description above does not specify a particular choice of $A_{\Psi_{\text{meta}}}, A_{\Psi_{\text{adapt}}}, \mathcal{L}_T, \mathcal{L}_V$. This is intentional, as there are many reasonable choices for these quantities. Because of this, we believe that ADKF-IFT should be considered a *general framework*, with a particular choice for these being an *instantiaton* of the ADKF-IFT framework. We give examples of this in Sections 3.3 and 3.4.

## 3.2 Efficient Meta-Training Algorithm

In general, optimizing bilevel optimization objectives such as Equation (1) is computationally complex, mainly because each evaluation of the objective requires solving a separate inner optimization problem (2). Although calculating the *hypergradient* (i.e., total derivative) of the validation loss $\mathcal{L}_V$ w.r.t. the meta-learned parameters $\boldsymbol{\psi}_{\text{meta}}$ would allow Equation (1) to be solved with gradient-based optimization:

$$\frac{d\mathcal{L}_V}{d\boldsymbol{\psi}_{\text{meta}}} = \frac{\partial\mathcal{L}_V}{\partial\boldsymbol{\psi}_{\text{meta}}} + \frac{\partial\mathcal{L}_V}{\partial\boldsymbol{\psi}^*_{\text{adapt}}} \frac{\partial\boldsymbol{\psi}^*_{\text{adapt}}}{\partial\boldsymbol{\psi}_{\text{meta}}}, \tag{4}$$

Equation (4) reveals that this requires calculating $\partial\boldsymbol{\psi}^*_{\text{adapt}}/\partial\boldsymbol{\psi}_{\text{meta}}$, i.e., how the optimal task-specific parameters $\boldsymbol{\psi}^*_{\text{adapt}}(\boldsymbol{\psi}_{\text{meta}}, \mathcal{S}_{\mathcal{T}})$ change with respect to the meta-learned parameters $\boldsymbol{\psi}_{\text{meta}}$. Calculating this naively with automatic differentiation platforms would require tracking the gradient through many iterations of the inner optimization (2), which in practice requires too much memory to be feasible. Fortunately, because $\boldsymbol{\psi}^*_{\text{adapt}}$ is an optimum of the train loss $\mathcal{L}_T$, Cauchy's *Implicit Function Theorem* (IFT) provides a formula for calculating $\partial\boldsymbol{\psi}^*_{\text{adapt}}/\partial\boldsymbol{\psi}_{\text{meta}}$ for an arbitrary value of the meta-learned parameters $\boldsymbol{\psi}'_{\text{meta}}$ and a given task $\mathcal{T}'$:

$$\left.\frac{\partial\boldsymbol{\psi}^*_{\text{adapt}}}{\partial\boldsymbol{\psi}_{\text{meta}}}\right|_{\boldsymbol{\psi}'_{\text{meta}}} = -\left(\frac{\partial^2\mathcal{L}_T(\boldsymbol{\psi}_{\text{meta}}, \boldsymbol{\psi}_{\text{adapt}}, \mathcal{S}_{\mathcal{T}'})}{\partial\boldsymbol{\psi}_{\text{adapt}}\partial\boldsymbol{\psi}^T_{\text{adapt}}}\right)^{-1} \left.\frac{\partial^2\mathcal{L}_T(\boldsymbol{\psi}_{\text{meta}}, \boldsymbol{\psi}_{\text{adapt}}, \mathcal{S}_{\mathcal{T}'})}{\partial\boldsymbol{\psi}_{\text{adapt}}\partial\boldsymbol{\psi}^T_{\text{meta}}}\right|_{\boldsymbol{\psi}'_{\text{meta}}, \boldsymbol{\psi}'_{\text{adapt}}}, \tag{5}$$

where $\boldsymbol{\psi}'_{\text{adapt}} = \boldsymbol{\psi}^*_{\text{adapt}}(\boldsymbol{\psi}'_{\text{meta}}, \mathcal{S}_{\mathcal{T}'})$. A full statement of the implicit function theorem in the context of ADKF-IFT can be found in Appendix A. The only potential problem with Equation (5) is the computation and inversion of the Hessian matrix $\partial^2\mathcal{L}_T(\boldsymbol{\psi}_{\text{meta}}, \boldsymbol{\psi}_{\text{adapt}}, \mathcal{S}_{\mathcal{T}})/\partial\boldsymbol{\psi}_{\text{adapt}}\partial\boldsymbol{\psi}^T_{\text{adapt}}$. This computation can be done exactly if $|A_{\Psi_{\text{adapt}}}|$ is small, which is the case considered in this paper (as will be discussed in Section 3.4). Otherwise, an approximation to the inverse Hessian (e.g., Neumann approximation (Lorraine et al., 2020; Clarke et al., 2022)) could be used, which reduces both the memory and computational complexities to $\mathcal{O}(|A_\Psi|)$. Combining Equations (4) and (5), we have a recipe for computing the hypergradient $d\mathcal{L}_V/d\boldsymbol{\psi}_{\text{meta}}$ exactly for a single task, as summarized in Algorithm 1. The meta-learned parameters $\boldsymbol{\psi}_{\text{meta}}$ can then be updated with the expected hypergradient over $p(\mathcal{T})$.

---

**Algorithm 1** Exact hypergradient computation in ADKF-IFT.

---

1: **Input:** a training task $\mathcal{T}'$ and the current meta-learned parameters $\boldsymbol{\psi}'_{\text{meta}}$.

2: Solve Equation (2) to obtain $\boldsymbol{\psi}'_{\text{adapt}} = \boldsymbol{\psi}^*_{\text{adapt}}(\boldsymbol{\psi}'_{\text{meta}}, \mathcal{S}_{\mathcal{T}'})$.

3: Compute $\mathbf{g}_1 = \left.\frac{\partial \mathcal{L}_V(\boldsymbol{\psi}_{\text{meta}}, \boldsymbol{\psi}_{\text{adapt}}, \mathcal{T}')}{\partial \boldsymbol{\psi}_{\text{meta}}}\right|_{\boldsymbol{\psi}'_{\text{meta}}, \boldsymbol{\psi}'_{\text{adapt}}}$ and $\mathbf{g}_2 = \left.\frac{\partial \mathcal{L}_V(\boldsymbol{\psi}_{\text{meta}}, \boldsymbol{\psi}_{\text{adapt}}, \mathcal{T}')}{\partial \boldsymbol{\psi}_{\text{adapt}}}\right|_{\boldsymbol{\psi}'_{\text{meta}}, \boldsymbol{\psi}'_{\text{adapt}}}$ by auto-diff.

4: Compute the Hessian $\mathbf{H} = \left.\frac{\partial^2 \mathcal{L}_T(\boldsymbol{\psi}_{\text{meta}}, \boldsymbol{\psi}_{\text{adapt}}, \mathcal{S}_{\mathcal{T}'})}{\partial \boldsymbol{\psi}_{\text{adapt}} \partial \boldsymbol{\psi}_{\text{adapt}}^T}\right|_{\boldsymbol{\psi}'_{\text{meta}}, \boldsymbol{\psi}'_{\text{adapt}}}$ by auto-diff.

5: Solve the linear system $\mathbf{v}\,\mathbf{H} = \mathbf{g}_2$ for $\mathbf{v}$.

6: Compute the mixed partial derivatives $\mathbf{P} = \left.\frac{\partial^2 \mathcal{L}_T(\boldsymbol{\psi}_{\text{meta}}, \boldsymbol{\psi}_{\text{adapt}}, \mathcal{S}_{\mathcal{T}'})}{\partial \boldsymbol{\psi}_{\text{adapt}} \partial \boldsymbol{\psi}_{\text{meta}}^T}\right|_{\boldsymbol{\psi}'_{\text{meta}}, \boldsymbol{\psi}'_{\text{adapt}}}$ by auto-diff.

7: **Output:** the hypergradient $\frac{d \mathcal{L}_V}{d \boldsymbol{\psi}_{\text{meta}}} = \mathbf{g}_1 - \mathbf{v}\,\mathbf{P}$. ▷ Equations (4) and (5)

---

### 3.3 ADKF-IFT AS A UNIFICATION OF PREVIOUS METHODS

In prior work, the most common method used to train deep kernel GPs is to minimize the negative log marginal likelihood (NLML) on a single dataset (optionally with extra regularization terms). This is commonly referred to as *Deep Kernel Learning* (DKL) (Wilson et al., 2016b), and is stated explicitly in Equation (6). The most notable departure from DKL is *Deep Kernel Transfer* (DKT) (Patacchiola et al., 2020), which instead proposes to train deep kernel GPs entirely using meta-learning, minimizing the *expected* NLML over a distribution of training tasks, as is stated explicitly in Equation (7).

$$\boldsymbol{\psi}^* = \arg\min_{\boldsymbol{\psi}} \text{NLML}\left(\boldsymbol{\psi}, \mathcal{S}_{\mathcal{T}}\right) \quad (6) \qquad \boldsymbol{\psi}^* = \arg\min_{\boldsymbol{\psi}} \mathbb{E}_{p(\mathcal{T})}[\text{NLML}(\boldsymbol{\psi}, \mathcal{T})] \quad (7)$$

Interestingly, both DKL and DKT can be viewed as *special cases* of the general ADKF-IFT framework. It is simple to see that choosing the partition to be $A_{\Psi_{\text{meta}}} = \varnothing$, $A_{\Psi_{\text{adapt}}} = A_{\Psi}$ and the train loss $\mathcal{L}_T$ to be the NLML in Equations (1) and (2) yields Equation (6): DKL is just ADKF-IFT if no parameters are meta-learned. Similarly, choosing the partition to be $A_{\Psi_{\text{meta}}} = A_{\Psi}$, $A_{\Psi_{\text{adapt}}} = \varnothing$ and the validation loss $\mathcal{L}_V$ to be the NLML in Equations (1) and (2) yields Equation (7): DKT is just ADKF-IFT if all parameters are meta-learned. This makes ADKF-IFT *strictly more general* than these two methods.

### 3.4 HIGHLIGHTED ADKF-IFT INSTANTIATION: META-LEARN $\phi$, ADAPT $\boldsymbol{\theta}$

Among the many possible variations of ADKF-IFT, we wish to highlight the following instantation:

- $A_{\Psi_{\text{meta}}} = A_\Phi$, i.e., all feature extractor parameters $\phi$ are meta-learned across tasks.
- $A_{\Psi_{\text{adapt}}} = A_\Theta$, i.e., all base kernel parameters $\boldsymbol{\theta}$ (e.g., noise, lengthscales, etc) are adapted.
- The train loss $\mathcal{L}_T$ and validation loss $\mathcal{L}_V$ are the negative log GP marginal likelihood on $\mathcal{S}_{\mathcal{T}}$ and the negative log joint GP predictive posterior on $\mathcal{Q}_{\mathcal{T}}$ given $\mathcal{S}_{\mathcal{T}}$, respectively. Please refer to Appendix B for equations of these loss functions.

There are several benefits to this choice. First, this particular choice of loss functions has the advantage that the prediction procedure during meta-testing (as defined in Equation (3)) exactly matches the meta-training procedure, thereby closely following the principle of *learning to learn*. Second, the partition of parameters can be intuitively understood as meta-learning a generally useful feature extractor $\mathbf{f}_\phi$ such that it is possible on average to fit a low-loss GP to the feature representations extracted by $\mathbf{f}_\phi$ for each individual task. This is very similar to previous transfer learning approaches. Third, since most GP base kernels have only a handful of parameters, the Hessian in Equation (5) can be computed and inverted exactly during meta-training using Algorithm 1; this removes any need for Hessian approximations. Fourth, the inner optimization (2) for $\boldsymbol{\psi}_{\text{adapt}}$ is computationally efficient, as it does not require backpropagating through the feature extractor $\mathbf{f}_\phi$.

More generally, we conjecture that adapting just the base kernel parameters will allow ADKF-IFT to achieve a better balance between overfitting and underfitting than either DKL or DKT. The relationship between these methods are visualized in Figure 1. Panel (c) shows DKL, which trains a separate deep kernel GP for each task. It is not hard to imagine that this can lead to severe overfitting for small datasets, which has been observed empirically by Ober et al. (2021). Panel (b) shows DKT,

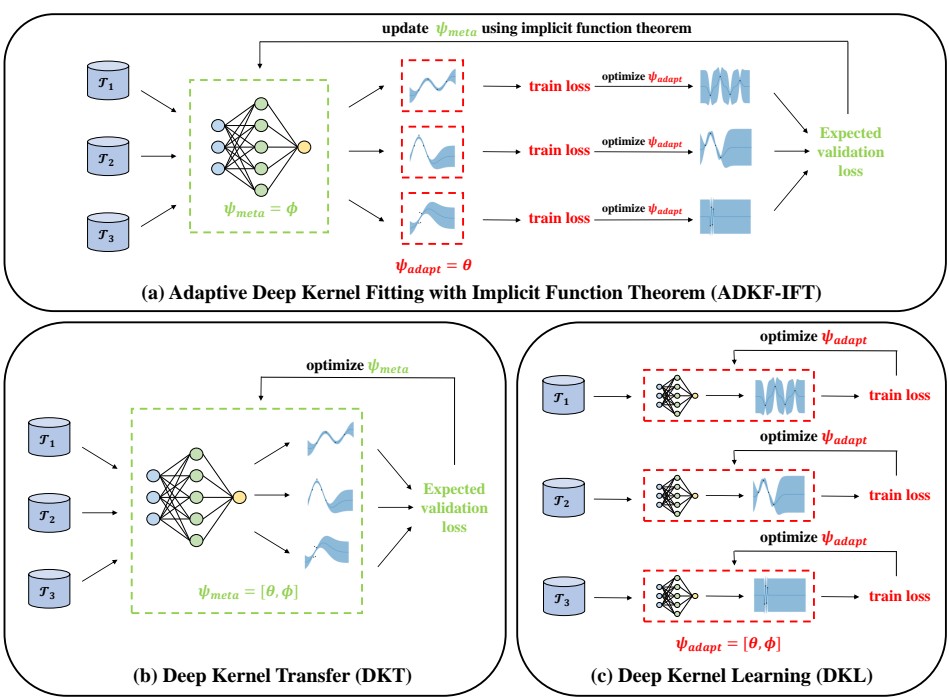

Figure 1: A contrastive diagram illustrating the training procedures of ADKF-IFT, DKT, and DKL.

which prevents overfitting by fitting one deep kernel GP for all tasks. However, this implicitly makes a strong assumption that all tasks come from an identical distribution over functions, including the same noise level, same amplitude, and same characteristic lengthscales, which is unlikely to hold in practice. Panel (a) shows ADKF-IFT, which allows these important parameters to be adapted, while still regularizing the feature extractor with meta-learning. We conjecture that adapting the base kernel parameters is more appropriate given the expected differences between tasks: two related tasks are more likely to have different noise levels or characteristic lengthscales than to require substantially different feature representations. We refer the readers to Appendix H for more discussions.

## 4    RELATED WORK

ADKF-IFT is part of a growing body of literature of techniques to train deep kernel GPs. As discussed in Section 3.3, ADKF-IFT *generalizes* DKL (Wilson et al., 2016b) and DKT (Patacchiola et al., 2020), which exclusively use single-task learning and meta-learning, respectively. Liu et al. (2020) and van Amersfoort et al. (2021) propose adding regularization terms to the loss of DKL in order to mitigate overfitting. These works are better viewed as *complementary* to ADKF-IFT rather than *alternatives*: their respective regularization terms could easily be added to $\mathcal{L}_T$ in Equation (2) to improve performance. However, the regularization strategies in both of these papers are designed for continuous inputs only, limiting their applicability to structured data like molecules.

ADKF-IFT can also be viewed as a meta-learning algorithm comparable to many previously-proposed methods (Lake et al., 2011; Vinyals et al., 2016; Garnelo et al., 2018; Triantafillou et al., 2019; Park & Oliva, 2019; Tian et al., 2020; Chen et al., 2021; Liu et al., 2021; Wistuba & Grabocka, 2021; Patacchiola et al., 2022). One distinguishing feature of ADKF-IFT is that it is specially designed for deep kernel GPs, whereas most methods from computer vision are designed exclusively for neural network models, which as previously stated are unsuitable when reliable uncertainty estimates are required. Furthermore, many of these algorithms such as ProtoNet (Snell et al., 2017) are designed principally or exclusively for classification, while ADKF-IFT is suited to both regression and classification. Compared to model-agnostic frameworks like MAML (Finn et al., 2017), ADKF-IFT does not require coarse approximations of the hypergradient due to its use of the implicit function theorem. For further discussions of related works, please refer to Appendix I.

Table 1: Mean test performance (AUROC%) with standard deviations of all compared methods on MoleculeNet benchmark tasks at support set size 20 (i.e., 2-way 10-shot).

| Method | MoleculeNet benchmark task (#compounds) | | | |
|---|---|---|---|---|
| | Tox21 (8,014) | SIDER (1,427) | MUV (93,127) | ToxCast (8,615) |
| Siamese | $80.40 \pm 0.35$ | $71.10 \pm 4.32$ | $59.59 \pm 5.13$ | - |
| ProtoNet | $74.98 \pm 0.32$ | $64.54 \pm 0.89$ | $65.88 \pm 4.11$ | $63.70 \pm 1.26$ |
| MAML | $80.21 \pm 0.24$ | $70.43 \pm 0.76$ | $63.90 \pm 2.28$ | $66.79 \pm 0.85$ |
| TPN | $76.05 \pm 0.24$ | $67.84 \pm 0.95$ | $65.22 \pm 5.82$ | $62.74 \pm 1.45$ |
| EGNN | $81.21 \pm 0.16$ | $72.87 \pm 0.73$ | $65.20 \pm 2.08$ | $63.65 \pm 1.57$ |
| IterRefLSTM | $81.10 \pm 0.17$ | $69.63 \pm 0.31$ | $45.56 \pm 5.12$ | - |
| PAR | $82.06 \pm 0.12$ | $\mathbf{74.68 \pm 0.31}$ | $66.48 \pm 2.12$ | $69.72 \pm 1.63$ |
| ADKF-IFT | $\mathbf{82.43 \pm 0.60}$ | $67.72 \pm 1.21$ | $\mathbf{98.18 \pm 3.05}$ | $\mathbf{72.07 \pm 0.81}$ |
| Pre-GNN | $82.14 \pm 0.08$ | $73.96 \pm 0.08$ | $67.14 \pm 1.58$ | $73.68 \pm 0.74$ |
| Meta-MGNN | $82.97 \pm 0.10$ | $75.43 \pm 0.21$ | $68.99 \pm 1.84$ | - |
| Pre-PAR | $84.93 \pm 0.11$ | $\mathbf{78.08 \pm 0.16}$ | $69.96 \pm 1.37$ | $75.12 \pm 0.84$ |
| Pre-ADKF-IFT | $\mathbf{86.06 \pm 0.35}$ | $70.95 \pm 0.60$ | $\mathbf{95.74 \pm 0.37}$ | $\mathbf{76.22 \pm 0.13}$ |

## 5 EXPERIMENTAL EVALUATION ON MOLECULES

In this section, we evaluate the empirical performance of ADKF-IFT from Section 3.4. We choose to focus our experiments exclusively on molecular property prediction and optimization tasks because we believe this application would benefit greatly from better GP models: firstly because many existing methods struggle on small datasets of size $\sim 10^2$ which are ubiquitous in chemistry, and secondly because many tasks in chemistry require high-quality uncertainty estimates. First, we evaluate ADKF-IFT on four commonly used benchmark tasks from MoleculeNet (Wu et al., 2018), finding that ADKF-IFT achieves state-of-the-art results on most tasks (Section 5.1). Second, we evaluate ADKF-IFT on the larger-scale FS-Mol benchmark (Stanley et al., 2021), finding that ADKF-IFT is the best-performing method (Section 5.2). In particular, our results support the hypothesis from Section 3.4 that ADKF-IFT achieves a better balance between overfitting and underfitting than DKL and DKT. Finally, we show that the ADKF-IFT feature representation is transferable to out-of-domain molecular property prediction and optimization tasks (Section 5.3). The general configurations of ADKF-IFT for all experiments considered in this paper are shown in Appendix B.

### 5.1 FEW-SHOT MOLECULAR PROPERTY PREDICTION ON THE MOLECULENET BENCHMARK

**Benchmark and Baselines.** We compare **ADKF-IFT** with two types of baselines on four few-shot molecular property classification benchmark tasks (Tox21, SIDER, MUV, and ToxCast) from MoleculeNet (Wu et al., 2018) (see Appendix C for more details of MoleculeNet benchmark tasks): 1) methods with feature extractor trained from scratch: **Siamese** (Koch, 2015), **ProtoNet** (Snell et al., 2017), **MAML** (Finn et al., 2017), **TPN** (Liu et al., 2019), **EGNN** (Kim et al., 2019), **IterRefLSTM** (Altae-Tran et al., 2017) and **PAR** (Wang et al., 2021); and 2) methods that fine-tune a pretrained feature extractor: **Pre-GNN** (Hu* et al., 2020), **Meta-MGNN** (Guo et al., 2021) and **Pre-PAR** (Wang et al., 2021). **Pre-ADKF-IFT** refers to ADKF-IFT starting from a pretrained feature extractor. All compared methods in this section use GIN (Xu et al., 2019) as their feature extractors. The pretrained weights for the methods of the second type are provided by Hu* et al. (2020).

**Evaluation Procedure.** We follow exactly the same evaluation procedure as that in Wang et al. (2021); Hu* et al. (2020); Guo et al. (2021). The task-level metric is AUROC (area under the receiver operating characteristic curve). We report the averaged performance over ten runs with different random seeds for each compared method at the support set size 20 (i.e., 2-way 10-shot, as the support sets in MoleculeNet are balanced). We did not perform 1-shot learning, as it is an unrealistic setting in real-world drug discovery tasks. All baseline results are taken from Wang et al. (2021).

**Performance.** Table 1 shows that ADKF-IFT and Pre-ADKF-IFT achieve the best performance on Tox21, MUV, and ToxCast. In general, the larger the dataset is, the larger the performance gains of our method over other baselines are, highlighting the scalability of our method. In particular, our method outperforms all baselines by a wide margin on MUV due to the relatively large amount of available compounds, but underperforms many baselines on SIDER due to a lack of compounds.

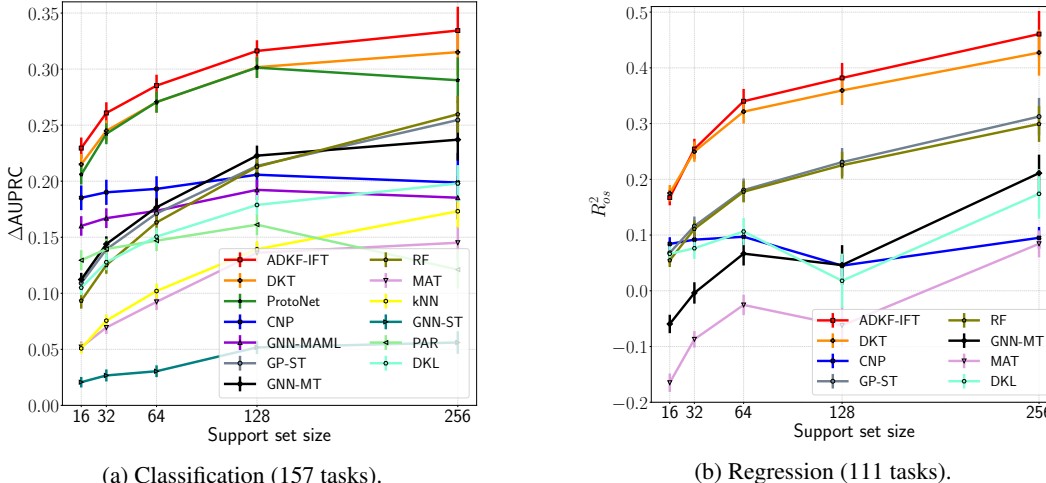

Figure 2: Mean performance with standard errors of all compared methods on all FS-Mol test tasks.

## 5.2 Few-shot Molecular Property Prediction on the FS-Mol Benchmark

**Benchmark.** We further conduct our evaluation on the FS-Mol benchmark (Stanley et al., 2021), which contains a carefully constructed set of few-shot learning tasks for molecular property prediction. FS-Mol contains over 5,000 tasks with 233,786 unique compounds from ChEMBL27 (Mendez et al., 2019), split into training (4,938 tasks), validation (40 tasks), and test (157 tasks) sets. Each task is associated with a protein target. The original benchmark only considers binary classification of active/inactive compounds, but we include the regression task (for the actual numeric activity target IC50 or EC50) in our evaluation as well, as it is a desired and more preferred task to do in real-world drug discovery projects.

**Baselines.** We compare **ADKF-IFT** with four categories of baselines: 1) single-task methods: Random Forest (**RF**), k-Nearest Neighbors (**kNN**), single-task GP with Tanimoto kernel (**GP-ST**) (Ralaivola et al., 2005), single-task GNN (**GNN-ST**) (Gilmer et al., 2017), Deep Kernel Learning (**DKL**) (Wilson et al., 2016b); 2) multi-task pretraining: multi-task GNN (**GNN-MT**) (Corso et al., 2020; Gilmer et al., 2017); 3) self-supervised pretraining: Molecule Attention Transformer (**MAT**) (Maziarka et al., 2020); 4) meta-learning methods: Property-Aware Relation Networks (**PAR**) (Wang et al., 2021), Prototypical Network with Mahalanobis distance (**ProtoNet**) (Snell et al., 2017), Model-Agnostic Meta-Learning (**GNN-MAML**) (Finn et al., 2017), Conditional Neural Process (**CNP**) (Garnelo et al., 2018), Deep Kernel Transfer (**DKT**) (Patacchiola et al., 2020). The GNN feature extractor architecture $\mathbf{f}_\phi$ used for DKL, PAR, CNP, DKT, and ADKF-IFT is the same as that used for ProtoNet, GNN-ST, GNN-MT, and GNN-MAML in Stanley et al. (2021). All multi-task and meta-learning methods are trained from scratch on FS-Mol training tasks. MAT is pretrained on 2 million molecules sampled from the ZINC15 dataset (Sterling & Irwin, 2015). The classification results for RF, kNN, GNN-ST, GNN-MT, MAT, ProtoNet, and GNN-MAML are reproduced according to Stanley et al. (2021). Detailed configurations of all compared methods can be found in Appendix D.

**Evaluation Procedure.** The task-level metrics for binary classification and regression are $\Delta$AUPRC (change in area under the precision-recall curve) and $R^2_{os}$ (predictive/out-of-sample coefficient of determination), respectively. Details of these metrics can be found in Appendix E. We follow exactly the same evaluation procedure as that in Stanley et al. (2021), where the averaged performance over ten different stratified support/query random splits of every test task is reported for each compared method. This evaluation process is performed for five different support set sizes 16, 32, 64, 128, and 256. Note that the support sets are generally unbalanced for the classification task in FS-Mol, which is natural as the majority of the candidate molecules are inactive in drug discovery.

**Overall Performance.** Figure 2 shows the overall test performance of all compared methods. Note that RF is a strong baseline method, as it is widely used in real-world drug discovery projects and has comparable performance to many pretraining methods. The results indicate that ADKF-IFT outperforms all the other compared methods at all considered support set sizes for the classification

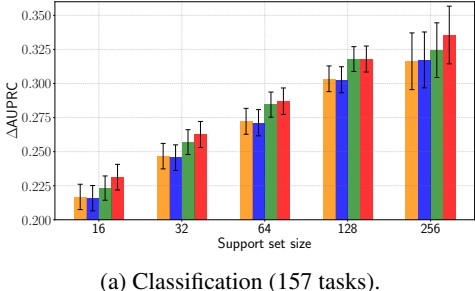
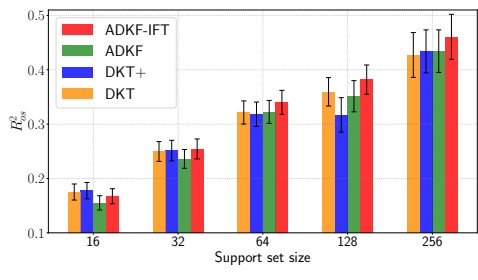

(a) Classification (157 tasks).
(b) Regression (111 tasks).

Figure 3: Mean performance with standard errors of ablation models on all FS-Mol test tasks. ADKF is like ADKF-IFT but assuming $\partial\theta^*/\partial\phi = \mathbf{0}$, i.e., updating $\phi$ with the direct gradient $\partial\mathcal{L}_V/\partial\phi$. DKT+ is like DKT but tuning the base kernel parameters $\theta$ during meta-testing.

task. For the regression task, the performance gains of ADKF-IFT over the second best method, namely DKT, get larger as the support set size increases. In Appendix F.1, we show that ADKF-IFT achieves the best mean rank for both classification and regression at all considered support set sizes.

**Statistical Comparison.** We perform two-sided Wilcoxon signed-rank tests (Wilcoxon, 1992) to compare the performance of ADKF-IFT and the next best method, namely DKT. The exact $p$-values from these statistical tests can be found in Appendix F.2. The results indicate that ADKF-IFT *significantly* outperforms DKT for the classification task at all considered support set sizes and for the regression task at support set sizes 64, 128, and 256 (at significance level $\alpha = 0.05$).

**Ablation Study.** To show that 1) the bilevel optimization objective for ADKF-IFT is essential for learning informative feature representations and 2) the performance gains of ADKF-IFT are not simply caused by tuning the base kernel parameters $\theta$ at meta-test time, we consider two ablation models: DKT+ and ADKF. The test performance of these models are shown in Figure 3. For ADKF, we follow the ADKF-IFT training scheme but assume $\partial\theta^*/\partial\phi = \mathbf{0}$, i.e., updating the feature extractor parameters $\phi$ with the *direct gradient* $\partial\mathcal{L}_V/\partial\phi$ rather than $d\mathcal{L}_V/d\phi$. The results show that ADKF consistently underperforms ADKF-IFT, indicating that the hypergradient for the bilevel optimization objective has non-negligible contributions to learning better feature representations. For DKT+, we take a model trained by DKT and adapt the base kernel parameters $\theta$ on each task at meta-test time. The results show that DKT+ does not improve upon DKT, indicating that tuning the base kernel parameters $\theta$ at meta-test time is not sufficient for obtaining better test performance with DKT.

**Sub-benchmark Performance.** The tasks in FS-Mol can be partitioned into 7 sub-benchmarks by Enzyme Commission number (Webb et al., 1992). In Appendix F.3, we show the test performance of top performing methods on each sub-benchmark. The results indicate that, in addition to achieving best overall performance, ADKF-IFT achieves the best performance on all sub-benchmarks for the regression task and on more than half of the sub-benchmarks for the classification task.

## 5.3 OUT-OF-DOMAIN MOLECULAR PROPERTY PREDICTION AND OPTIMIZATION

Finally, we demonstrate that the feature representation learned by ADKF-IFT is useful not only for in-domain molecular property prediction tasks but also for out-of-domain molecular property prediction and optimization tasks. For this, we perform experiments involving finding molecules with best desired target properties within given out-of-domain datasets using Bayesian optimization (BO) with a GP surrogate model operating on top of compared feature representations. We use the expected improvement acquisition function (Jones et al., 1998) with query-batch size 1. All compared feature representations are extracted using the models trained on the FS-Mol dataset from scratch in Section 5.2, except for the pretrained MAT representation and fingerprint. We compare them on four representative molecular design tasks outside of FS-Mol. Detailed configuration of the GP and descriptions of the tasks can be found in Appendix G. We repeat each BO experiment 20 times, each time starting from 16 randomly sampled molecules from the worst $\sim 700$ molecules within the dataset. Figure 4 shows that the ADKF-IFT representation enables fastest discovery of top performing molecules for the molecular docking, antibiotic discovery, and material design tasks. For

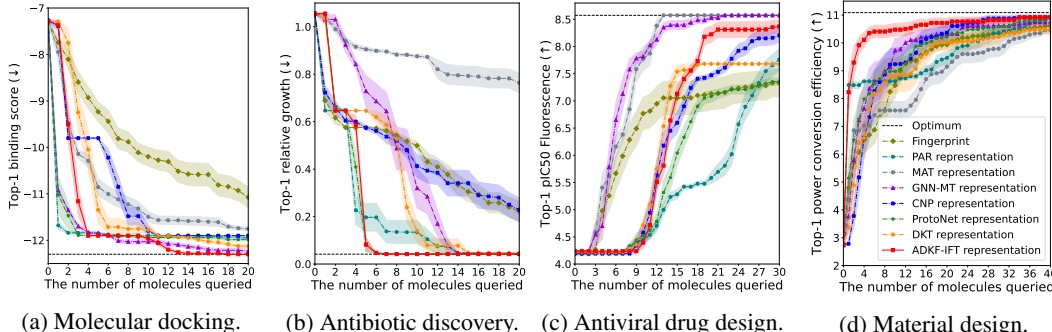

| (a) Molecular docking. | (b) Antibiotic discovery. | (c) Antiviral drug design. | (d) Material design. |

Figure 4: Mean top-1 target values with standard errors as a function of the number of molecules queried for all compared feature representations on four out-of-domain molecular optimization tasks.

Table 2: Mean predictive performance (test NLL) with standard errors of a GP operating on top of each compared feature representation on the four out-of-domain molecular design tasks.

| Feature representation | Out-of-domain molecular design task | | | |
| --- | --- | --- | --- | --- |
| | Molecular docking | Antibiotic discovery | Antiviral drug design | Material design |
| Fingerprint | $1.138 \pm 0.014$ | $1.669 \pm 0.075$ | $\mathbf{4.601 \pm 0.086}$ | $1.091 \pm 0.011$ |
| PAR | $1.270 \pm 0.019$ | $2.185 \pm 0.115$ | $4.840 \pm 0.086$ | $1.283 \pm 0.017$ |
| MAT | $1.528 \pm 0.028$ | $2.390 \pm 0.104$ | $4.797 \pm 0.088$ | $2.198 \pm 0.063$ |
| GNN-MT | $1.994 \pm 0.050$ | $3.692 \pm 0.225$ | $6.399 \pm 0.181$ | $7.254 \pm 0.217$ |
| CNP | $1.493 \pm 0.028$ | $2.537 \pm 0.162$ | $5.005 \pm 0.086$ | $1.741 \pm 0.043$ |
| ProtoNet | $1.147 \pm 0.013$ | $1.615 \pm 0.094$ | $5.060 \pm 0.086$ | $1.032 \pm 0.009$ |
| DKT | $1.167 \pm 0.012$ | $1.602 \pm 0.073$ | $4.975 \pm 0.092$ | $1.026 \pm 0.009$ |
| ADKF-IFT | $\mathbf{1.137 \pm 0.011}$ | $\mathbf{1.496 \pm 0.043}$ | $4.781 \pm 0.087$ | $\mathbf{0.996 \pm 0.007}$ |

the antiviral drug design task, although the ADKF-IFT representation underperforms the MAT and GNN-MT representations, it still achieves competitive performance compared to other baselines.

Table 2 explicitly reports the regression predictive performance of a GP operating on top of each compared feature representation for these four out-of-domain molecular design tasks. The configuration of the GP is the same as that in the BO experiments. We report test negative log likelihood (NLL) averaged over 200 support/query random splits (100 for each of the support set sizes 32 and 64). The results show that the ADKF-IFT representation has the best test NLL on the molecular docking, antibiotic discovery, and material design tasks, and ranks second on the antiviral drug design task.

## 6 CONCLUSION

We have proposed Adaptive Deep Kernel Fitting with Implicit Function Theorem (ADKF-IFT), a novel framework for fitting deep kernels that interpolates between meta-learning and conventional deep kernel learning. ADKF-IFT meta-learns a feature extractor across tasks such that the task-specific GP models estimated on top of the extracted feature representations can achieve the lowest possible prediction error on average. ADKF-IFT is implemented by solving a bilevel optimization objective via implicit differentiation. We have shown that ADKF-IFT is a unifying framework containing DKL and DKT as special cases. We have demonstrated that ADKF-IFT learns generally useful feature representations, achieving state-of-the-art performance on a variety of real-world few-shot molecular property prediction tasks and on out-of-domain molecular property prediction and optimization tasks. We believe that ADKF-IFT could potentially be an important method to produce well-calibrated models for fully-automated high-throughput experimentation in the future.

ACKNOWLEDGMENTS AND DISCLOSURE OF FUNDING

We thank Massimiliano Patacchiola, John Bronskill, Marcin Sendera, and Richard E. Turner for helpful discussions and feedback. WC acknowledges funding via a Cambridge Trust Scholarship (supported by the Cambridge Trust) and a Cambridge University Engineering Department Studentship (under grant G105682 NMZR/089 supported by Huawei R&D UK). AT acknowledges funding via a C T Taylor Cambridge International Scholarship. JMHL acknowledges support from a Turing AI Fellowship under grant EP/V023756/1.

ETHICS STATEMENT

We believe that the ethical implications of this work are minimal: this research involves no human subjects, no sensitive data where privacy is a concern, no domains where discrimination/bias/fairness is concerning, and is unlikely to have a noticeable social impact. Optimistically, our hope is that in the future ADKF-IFT could be used in the drug discovery pipeline to create new beneficial medicines, giving it an overall *positive* ethical impact. However, as with most research in machine learning, new modelling techniques could be used by bad actors to cause harm more effectively, but we do not see how ADKF-IFT is more concerning than any other method in this regard.

REPRODUCIBILITY STATEMENT

Our implementation and experimental results can be found at: `https://github.com/Wenlin-Chen/ADKF-IFT`, which is based on a forked from FS-Mol (Stanley et al., 2021) and PAR (Wang et al., 2021). Details for the setup of ADKF-IFT can be found in Appendix B, while details for other FS-Mol baselines can be found in Appendix D and details for other MoleculeNet baselines can be found in Wang et al. (2021, Appendix B).

The arXiv version of this paper can be found at: `https://arxiv.org/abs/2205.02708`, which may be updated as needed.

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

## A  CAUCHY'S IMPLICIT FUNCTION THEOREM

We state Cauchy's Implicit Function Theorem (IFT) in the context of ADKF-IFT in Theorem 1.

**Theorem 1 (Implicit Function Theorem (IFT))** *Let $\mathcal{T}'$ be any given task. Suppose for some $\psi'_{meta}$ and $\psi'_{adapt}$ that $\left.\frac{\partial \mathcal{L}_T(\psi_{meta}, \psi_{adapt}, \mathcal{S}_{\mathcal{T}'})}{\partial \psi_{adapt}}\right|_{\psi'_{meta}, \psi'_{adapt}} = \mathbf{0}$. Suppose that $\frac{\partial \mathcal{L}_T}{\partial \psi_{adapt}}(\psi_{meta}, \psi_{adapt}, \mathcal{S}_{\mathcal{T}'})$ : $\Psi_{meta} \times \Psi_{adapt} \to \Psi_{adapt}$ is a continuously differentiable function w.r.t. $\psi_{meta}$ and $\psi_{adapt}$, and the Hessian $\left.\frac{\partial^2 \mathcal{L}_T(\psi_{meta}, \psi_{adapt}, \mathcal{S}_{\mathcal{T}'})}{\partial \psi_{adapt} \partial \psi_{adapt}^T}\right|_{\psi'_{meta}, \psi'_{adapt}}$ is invertible. Then, there exists an open set $U \in \Psi_{meta}$ containing $\psi'_{meta}$ and a function $\psi^*_{adapt}(\psi_{meta}, \mathcal{S}_{\mathcal{T}'}) : \Psi_{meta} \to \Psi_{adapt}$, such that $\psi'_{adapt} = \psi^*_{adapt}(\psi'_{meta}, \mathcal{S}_{\mathcal{T}'})$ and $\left.\frac{\partial \mathcal{L}_T(\psi_{meta}, \psi_{adapt}, \mathcal{S}_{\mathcal{T}'})}{\partial \psi_{adapt}}\right|_{\psi''_{meta}, \psi^*_{adapt}(\psi''_{meta}, \mathcal{S}_{\mathcal{T}'})} = \mathbf{0}, \forall \psi''_{meta} \in U$. Moreover, the rate at which $\psi^*_{adapt}(\psi_{meta}, \mathcal{S}_{\mathcal{T}'})$ is changing w.r.t. $\psi_{meta}$ for any $\psi''_{meta} \in U$ is given by*

$$
\left.\frac{\partial \psi^*_{adapt}(\psi_{meta}, \mathcal{S}_{\mathcal{T}'})}{\partial \psi_{meta}}\right|_{\psi''_{meta}}
$$

$$
= -\left(\frac{\partial^2 \mathcal{L}_T(\psi_{meta}, \psi_{adapt}, \mathcal{S}_{\mathcal{T}'})}{\partial \psi_{adapt} \partial \psi_{adapt}^T}\right)^{-1} \left.\frac{\partial^2 \mathcal{L}_T(\psi_{meta}, \psi_{adapt}, \mathcal{S}_{\mathcal{T}'})}{\partial \psi_{adapt} \partial \psi_{meta}^T}\right|_{\psi''_{meta}, \psi^*_{adapt}(\psi''_{meta}, \mathcal{S}_{\mathcal{T}'})} .
$$

## B  GENERAL CONFIGURATIONS OF ADKF-IFT FOR FEW-SHOT LEARNING EXPERIMENTS

In this paper, we consider the specific instantiation of ADKF-IFT from Section 3.4. Specifically, we set $A_{\Psi_{adapt}} = A_\Theta$ and $A_{\Psi_{meta}} = A_\Phi$, i.e., to meta-learn the feature extractor parameters $\phi$ across tasks and to adapt the base kernel parameters $\theta$ for each individual task. We choose the train loss $\mathcal{L}_T$ to be the negative log GP marginal likelihood evaluated on the support set $\mathcal{S}_\mathcal{T}$, as is common practice for choosing GP base kernel parameters:

$$
\begin{aligned}
\mathcal{L}_T(\psi_{\text{meta}}, \psi_{\text{adapt}}, \mathcal{S}_\mathcal{T}) &= -\log p(\mathcal{S}_\mathcal{T}^y \mid \mathcal{S}_\mathcal{T}^{\mathbf{x}}, \psi_{\text{meta}}, \psi_{\text{adapt}}) \\
&= \frac{1}{2}\left\langle \mathcal{S}_\mathcal{T}^y, \mathbf{K}_{\mathcal{S}_\mathcal{T}}^{-1} \mathcal{S}_\mathcal{T}^y \right\rangle + \frac{1}{2}\log\det(\mathbf{K}_{\mathcal{S}_\mathcal{T}}) + \frac{N_{\mathcal{S}_\mathcal{T}}}{2}\log(2\pi),
\end{aligned} \tag{8}
$$

where $\mathbf{K}_{\mathcal{S}_\mathcal{T}} = k_{\psi_{\text{meta}}, \psi_{\text{adapt}}}(\mathcal{S}_\mathcal{T}^{\mathbf{x}}, \mathcal{S}_\mathcal{T}^{\mathbf{x}}) + \sigma^2 \mathbf{I}_{N_{\mathcal{S}_\mathcal{T}}}$. We choose the validation loss $\mathcal{L}_V$ to be the negative log joint GP predictive posterior evaluated on the query set $\mathcal{Q}_\mathcal{T}$ given the support set $\mathcal{S}_\mathcal{T}$, also due to its common usage for making predictions with GPs:

$$
\begin{aligned}
\mathcal{L}_V(\psi_{\text{meta}}, \psi_{\text{adapt}}, \mathcal{T}) &= -\log p(\mathcal{Q}_\mathcal{T}^y \mid \mathcal{Q}_\mathcal{T}^{\mathbf{x}}, \mathcal{S}_\mathcal{T}, \psi_{\text{meta}}, \psi_{\text{adapt}}) \\
&= -\log\mathcal{N}(\mathcal{Q}_\mathcal{T}^y; \mathbf{K}_{\mathcal{Q}_\mathcal{T}\mathcal{S}_\mathcal{T}}\mathbf{K}_{\mathcal{S}_\mathcal{T}}^{-1}\mathcal{S}_\mathcal{T}^y, \mathbf{K}_{\mathcal{Q}_\mathcal{T}} - \mathbf{K}_{\mathcal{Q}_\mathcal{T}\mathcal{S}_\mathcal{T}}\mathbf{K}_{\mathcal{S}_\mathcal{T}}^{-1}\mathbf{K}_{\mathcal{S}_\mathcal{T}\mathcal{Q}_\mathcal{T}}),
\end{aligned} \tag{9}
$$

where $\mathbf{K}_{\mathcal{Q}_\mathcal{T}} = k_{\psi_{\text{meta}}, \psi_{\text{adapt}}}(\mathcal{Q}_\mathcal{T}^{\mathbf{x}}, \mathcal{Q}_\mathcal{T}^{\mathbf{x}}) + \sigma^2 \mathbf{I}_{N_{\mathcal{Q}_\mathcal{T}}}$ and $\mathbf{K}_{\mathcal{S}_\mathcal{T}\mathcal{Q}_\mathcal{T}} = \mathbf{K}_{\mathcal{Q}_\mathcal{T}\mathcal{S}_\mathcal{T}}^T = k_{\psi_{\text{meta}}, \psi_{\text{adapt}}}(\mathcal{S}_\mathcal{T}^{\mathbf{x}}, \mathcal{Q}_\mathcal{T}^{\mathbf{x}})$.

We solve the inner optimization problem (2) using the L-BFGS optimizer (Liu & Nocedal, 1989), since L-BFGS is the default choice for optimizing base kernel parameters in the GP literature. For the outer optimization problem (1), we approximate the expected hypergradient over $p(\mathcal{T})$ by averaging the hypergradients for a batch of $K$ randomly sampled training tasks at each step, and update the meta-learned parameters $\psi_{\text{meta}}$ with the averaged hypergradient using the Adam optimizer (Kingma & Ba, 2014) with learning rate $10^{-3}$ for MoleculeNet and $10^{-4}$ for FS-Mol. We set $K = 10$ for MoleculeNet and $K = 16$ for FS-Mol. For all experiments on FS-Mol, we evaluate the performance of our model on a small set of validation tasks during meta-training and use early stopping (Prechelt, 1998) to avoid overfitting of $\psi_{\text{meta}}$.

We use zero mean function and set Matérn52 without automatic relevance determination (ARD) (Neal, 1996) as the base kernel in ADKF-IFT, since the typical sizes of the support sets in few-shot learning are too small to adjust a relatively large number of ARD lengthscales in ADKF-IFT. The lengthscale in the base kernel of ADKF-IFT is initialized using the median heuristic (Garreau et al., 2017) for each task, with a log-normal prior centered at the initialization. Following Patacchiola et al. (2020), we treat binary classification as $\pm 1$ label regression for ADKF-IFT.

Table 3: Statistics of four few-shot molecular property prediction benchmarks from MoleculeNet.

| Statistic | MoleculeNet benchmark task | | | |
| --- | --- | --- | --- | --- |
| | Tox21 | SIDER | MUV | ToxCast |
| #compounds | 8,014 | 1,427 | 93,127 | 8,615 |
| #tasks | 12 | 27 | 17 | 617 |
| #training tasks | 9 | 21 | 12 | 450 |
| #test tasks | 3 | 6 | 5 | 167 |

## C  DETAILS OF MOLECULENET BENCHMARK TASKS

In Table 3, we summarize the four few-shot molecular property classification benchmark tasks (Tox21, SIDER, MUV, and ToxCast) from MoleculeNet (Wu et al., 2018) considered in Section 5.1.

## D  DETAILED CONFIGURATIONS OF ALL COMPARED METHODS FOR FS-MOL

**Single-task Methods.** Single-task methods (RF, kNN, GP-ST, GNN-ST, and DKL) are trained separately on the support set of each test task, without leveraging the knowledge contained in the training tasks. The implementations of RF, kNN, and GNN-ST are taken from Stanley et al. (2021). RF, kNN, and GP-ST operates on top of manually curated features obtained using RDKit. RF and kNN use extended connectivity fingerprint (Rogers & Hahn, 2010) (count-based fingerprint with radius 2 and size 2,048) and phys-chem descriptors (with size 42). GP-ST uses fingerprint (with radius 2 and 2,048 bits based on count simulation). DKL operates on top of a combination of extended connectivity fingerprint (Rogers & Hahn, 2010) (count-based fingerprint with radius 2 and size 2,048) and features extracted by a GNN. The base kernel used in DKL is the same as that used in ADKF-IFT. DKL is trained for 50 epochs on the support set of each test task. Hyperparameter search configurations for these methods are based on the extensive industrial experience from the authors of Stanley et al. (2021). GNN-ST uses a GNN with a hidden dimension of 128 and a gated readout function Gilmer et al. (2017), considering $\sim 30$ hyperparameter search configurations.

**Multi-task Pretraining.** The implementation of GNN-MT is taken from Stanley et al. (2021). GNN-MT shares a GNN with a hidden dimension of 128 using principal neighborhood message aggregation (Corso et al., 2020) across tasks, and uses a task-specific gated readout function Gilmer et al. (2017) and an MLP with one hidden layer on top for each individual task. The model is trained on the support sets of all training tasks with early stopping based on the validation performance on the validation tasks. The task-specific components of the model are fine-tuned for each test task.

**Self-supervised Pretraining.** The implementation of MAT is taken from Stanley et al. (2021). We use the official pretrained model parameters (Maziarka et al., 2020), which is pretrained on 2 million molecules sampled from the ZINC15 dataset (Sterling & Irwin, 2015). We fine-tuned it for each test task with hyperparameter search and early stopping based on 20% of the support set for each task.

**Meta-learning Methods.** Meta-learning methods (PAR, ProtoNet, GNN-MAML, CNP, DKT, and ADKF-IFT) enable knowledge transfer among related small datasets. The implementations of ProtoNet and GNN-MAML are taken from Stanley et al. (2021). The implementation of PAR is taken from its official implementation and integrated into the FS-Mol training and evaluation pipeline. PAR, ProtoNet, CNP, DKT, and ADKF-IFT operate on top of a combination of extended connectivity fingerprint (Rogers & Hahn, 2010) (count-based fingerprint with radius 2 and size 2,048) and features extracted by a GNN. The GNN feature extractor architecture used for DKL, PAR, CNP, DKT, and ADKF-IFT is the same as that used for ProtoNet, GNN-MAML, GNN-ST, and GNN-MT in Stanley et al. (2021), with the size of the feature representation being tuned on the validation tasks. The base kernel used in DKT is the same as that used in ADKF-IFT.

## E  TASK-LEVEL EVALUATION METRICS FOR FS-MOL

**Binary Classification.** Following Stanley et al. (2021), the task-level metric used for the binary classification task in FS-Mol is change in area under the precision-recall curve ($\Delta$AUPRC), which

Table 4: Mean ranks of all compared methods in terms of their performance on all FS-Mol test tasks.

(a) Classification (157 tasks).

| Method | Support set size | | | | |
|---|---|---|---|---|---|
| | 16 | 32 | 64 | 128 | 256 |
| GNN-ST | 11.29 | 11.53 | 11.75 | 11.85 | 12.19 |
| kNN | 10.89 | 10.48 | 10.33 | 10.15 | 9.37 |
| MAT | 10.43 | 10.44 | 10.19 | 9.69 | 9.70 |
| RF | 8.15 | 7.89 | 7.06 | 6.25 | 4.47 |
| PAR | 7.70 | 7.98 | 8.30 | 8.83 | 10.81 |
| GNN-MT | 7.33 | 7.18 | 7.08 | 6.59 | 6.53 |
| DKL | 7.28 | 7.49 | 7.98 | 8.42 | 8.21 |
| GP-ST | 6.71 | 6.57 | 6.28 | 6.18 | 5.14 |
| GNN-MAML | 6.36 | 6.92 | 7.42 | 7.89 | 8.90 |
| CNP | 5.00 | 5.81 | 6.36 | 6.91 | 7.78 |
| ProtoNet | 4.00 | 3.40 | 3.11 | 2.98 | 3.85 |
| DKT | 3.44 | 3.19 | 2.99 | 2.99 | 2.67 |
| ADKF-IFT | **2.41** | **2.12** | **2.14** | **2.26** | **1.38** |

(b) Regression (111 tasks).

| Method | Support set size | | | | |
|---|---|---|---|---|---|
| | 16 | 32 | 64 | 128 | 256 |
| MAT | 7.60 | 7.45 | 7.26 | 7.06 | 7.19 |
| GNN-MT | 6.61 | 6.40 | 6.15 | 5.95 | 5.58 |
| RF | 5.00 | 4.47 | 4.16 | 3.72 | 3.56 |
| DKL | 4.42 | 5.16 | 5.63 | 6.10 | 6.35 |
| GP-ST | 4.23 | 4.14 | 3.87 | 3.37 | 3.07 |
| CNP | 3.88 | 4.45 | 4.95 | 5.73 | 6.47 |
| DKT | **2.12** | 2.08 | 2.29 | 2.32 | 2.43 |
| ADKF-IFT | **2.12** | **1.86** | **1.68** | **1.74** | **1.36** |

is sensitive to the balance of the two classes in the query sets and allows for a comparison to the performance of a random classifier:

$$\Delta\text{AUPRC}(\text{target classifier}, \mathcal{T}) = \text{AUPRC}(\text{target classifier}, \mathcal{Q}_\mathcal{T}) - \text{AUPRC}(\text{random classifier}, \mathcal{Q}_\mathcal{T})$$
$$= \text{AUPRC}(\text{target classifier}, \mathcal{Q}_\mathcal{T}) - \frac{\#\text{positive data points in } \mathcal{Q}_\mathcal{T}}{N_{\mathcal{Q}_\mathcal{T}}}.$$

**Regression.** We propose to use the predictive/out-of-sample coefficient of determination ($R^2_{os}$) as the task-level metric for the regression task in FS-Mol, which takes into account forecast errors:

$$R^2_{os}(\text{target regressor } g, \mathcal{T}) = 1 - \frac{\sum_{(\mathbf{x}_m, y_m) \in \mathcal{Q}_\mathcal{T}} (y_m - g(\mathbf{x}_m))^2}{\sum_{y_m \in \mathcal{Q}^y_\mathcal{T}} (y_m - \bar{y}_{\mathcal{S}_\mathcal{T}})^2},$$

where $\bar{y}_{\mathcal{S}_\mathcal{T}} = \frac{1}{N_{\mathcal{S}_\mathcal{T}}} \sum_{y_n \in \mathcal{S}^y_\mathcal{T}} y_n$ is the mean target value in the support set $\mathcal{S}_\mathcal{T}$. This is different from the regular coefficient of determination ($R^2$), wherein the total sum of squares in the denominator are computed using the mean target value $\bar{y}_{\mathcal{Q}_\mathcal{T}} = \frac{1}{N_{\mathcal{Q}_\mathcal{T}}} \sum_{y_m \in \mathcal{Q}^y_\mathcal{T}} y_m$ in the query set $\mathcal{Q}_\mathcal{T}$.

# F FURTHER EXPERIMENTAL RESULTS ON FS-MOL

## F.1 OVERALL PERFORMANCE

Table 4 shows that ADKF-IFT achieves the best mean rank for both classification and regression tasks at all considered support set sizes. The trends of these mean ranks are consistent to those in Figure 2. Figures 6 and 7 show the box plots for the classification and regression performances of all compared methods on all FS-Mol test tasks, respectively. These plots are a disaggregated representation of the results in Figure 2.

## F.2 STATISTICAL COMPARISON

Table 5 shows the $p$-values from the two-sided Wilcoxon signed-rank test for statistical comparisons between ADKF-IFT and the next second method, namely DKT. The test results indicate that their median performance difference is nonzero (i.e., ADKF-IFT significantly outperforms DKT) for the classification task at all considered support set sizes and for the regression task at support set sizes 64, 128, and 256 (at significance level $\alpha = 0.05$). The $p$-values for statistical comparisons between ADKF-IFT and the two ablation models DKT+ and ADKF are also shown in Table 5, demonstrating that ADKF-IFT significantly outperforms these ablation models in most cases.

## F.3 SUB-BENCHMARK PERFORMANCE

The tasks in FS-Mol can be partitioned into 7 sub-benchmarks by Enzyme Commission (EC) number (Webb et al., 1992), which enables sub-benchmark evaluation within the entire benchmark. Ideally,

Table 5: $p$-values from the two-sided Wilcoxon signed-rank test for statistical comparisons between ADKF-IFT and DKT/DKT+/ADKF. The null hypothesis is that the median of their performance differences on all FS-Mol test tasks is zero. The significance level is set to $\alpha = 0.05$.

| Compared models | Task type | Support set size | | | | |
|---|---|---|---|---|---|---|
| | | 16 | 32 | 64 | 128 | 256 |
| ADKF-IFT vs DKT | Classification | $\mathbf{1.4 \times 10^{-12}}$ | $\mathbf{8.1 \times 10^{-14}}$ | $\mathbf{2.3 \times 10^{-12}}$ | $\mathbf{1.0 \times 10^{-8}}$ | $\mathbf{3.4 \times 10^{-7}}$ |
| | Regression | $8.2 \times 10^{-2}$ | $9.6 \times 10^{-2}$ | $\mathbf{3.7 \times 10^{-5}}$ | $\mathbf{7.1 \times 10^{-5}}$ | $\mathbf{9.8 \times 10^{-7}}$ |
| ADKF-IFT vs DKT+ | Classification | $\mathbf{3.2 \times 10^{-13}}$ | $\mathbf{7.0 \times 10^{-15}}$ | $\mathbf{2.3 \times 10^{-13}}$ | $\mathbf{1.2 \times 10^{-9}}$ | $\mathbf{1.6 \times 10^{-6}}$ |
| | Regression | $\mathbf{3.2 \times 10^{-2}}$ | $4.2 \times 10^{-1}$ | $\mathbf{3.4 \times 10^{-5}}$ | $\mathbf{5.2 \times 10^{-10}}$ | $\mathbf{1.2 \times 10^{-5}}$ |
| ADKF-IFT vs ADKF | Classification | $\mathbf{1.7 \times 10^{-2}}$ | $1.1 \times 10^{-1}$ | $4.8 \times 10^{-1}$ | $8.3 \times 10^{-1}$ | $\mathbf{1.6 \times 10^{-3}}$ |
| | Regression | $\mathbf{2.8 \times 10^{-3}}$ | $\mathbf{4.2 \times 10^{-4}}$ | $\mathbf{1.3 \times 10^{-3}}$ | $\mathbf{4.1 \times 10^{-6}}$ | $\mathbf{1.3 \times 10^{-5}}$ |

Table 6: Mean performance with standard errors of top performing methods on FS-Mol test tasks within each sub-benchmark (broken down by EC category) at support set size 64 (the median of all considered support sizes). Note that class 2 is most common in the FS-Mol training set ($\sim 1,500$ training tasks), whereas classes 6 and 7 are least common in the FS-Mol training set ($< 50$ training tasks each).

(a) Classification ($\Delta$AUPRC).

| FS-Mol sub-benchmark (EC category) | | | Method | | | | |
|---|---|---|---|---|---|---|---|
| Class | Description | #tasks | RF | GP-ST | ProtoNet | DKT | ADKF-IFT |
| 1 | oxidoreductases | 7 | $0.156 \pm 0.044$ | $0.152 \pm 0.040$ | $0.137 \pm 0.037$ | $0.145 \pm 0.040$ | $\mathbf{0.160 \pm 0.045}$ |
| 2 | kinases | 125 | $0.152 \pm 0.009$ | $0.161 \pm 0.009$ | $0.285 \pm 0.010$ | $0.282 \pm 0.010$ | $\mathbf{0.299 \pm 0.010}$ |
| 3 | hydrolases | 20 | $0.229 \pm 0.032$ | $0.230 \pm 0.032$ | $0.245 \pm 0.034$ | $0.254 \pm 0.034$ | $\mathbf{0.262 \pm 0.033}$ |
| 4 | lysases | 2 | $0.276 \pm 0.182$ | $\mathbf{0.284 \pm 0.189}$ | $0.265 \pm 0.211$ | $0.272 \pm 0.206$ | $0.279 \pm 0.201$ |
| 5 | isomerases | 1 | $0.166 \pm 0.040$ | $\mathbf{0.212 \pm 0.052}$ | $0.172 \pm 0.044$ | $0.204 \pm 0.058$ | $0.198 \pm 0.046$ |
| 6 | ligases | 1 | $0.149 \pm 0.035$ | $0.199 \pm 0.028$ | $0.170 \pm 0.028$ | $0.229 \pm 0.013$ | $\mathbf{0.231 \pm 0.022}$ |
| 7 | translocases | 1 | $\mathbf{0.128 \pm 0.039}$ | $0.109 \pm 0.049$ | $0.099 \pm 0.028$ | $0.122 \pm 0.022$ | $0.109 \pm 0.033$ |
| | all enzymes | 157 | $0.163 \pm 0.009$ | $0.171 \pm 0.009$ | $0.271 \pm 0.009$ | $0.271 \pm 0.010$ | $\mathbf{0.285 \pm 0.010}$ |

(b) Regression ($R^2_{os}$).

| FS-Mol sub-benchmark (EC category) | | | Method | | | | |
|---|---|---|---|---|---|---|---|
| Class | Description | #tasks | RF | GP-ST | CNP | DKT | ADKF-IFT |
| 1 | oxidoreductases | 6 | $0.108 \pm 0.087$ | $0.103 \pm 0.076$ | $-0.012 \pm 0.011$ | $0.098 \pm 0.078$ | $\mathbf{0.116 \pm 0.079}$ |
| 2 | kinases | 82 | $0.160 \pm 0.019$ | $0.162 \pm 0.022$ | $0.127 \pm 0.017$ | $0.343 \pm 0.022$ | $\mathbf{0.363 \pm 0.024}$ |
| 3 | hydrolases | 19 | $0.256 \pm 0.058$ | $0.267 \pm 0.061$ | $0.014 \pm 0.015$ | $0.295 \pm 0.063$ | $\mathbf{0.310 \pm 0.062}$ |
| 4 | lysases | 2 | $0.418 \pm 0.405$ | $0.417 \pm 0.416$ | $0.100 \pm 0.068$ | $0.440 \pm 0.418$ | $\mathbf{0.442 \pm 0.403}$ |
| 5 | isomerases | 1 | $0.125 \pm 0.077$ | $0.086 \pm 0.082$ | $-0.012 \pm 0.010$ | $0.209 \pm 0.113$ | $\mathbf{0.226 \pm 0.063}$ |
| 6 | ligases | 1 | $0.182 \pm 0.040$ | $0.202 \pm 0.079$ | $0.002 \pm 0.004$ | $0.277 \pm 0.035$ | $\mathbf{0.279 \pm 0.043}$ |
| | all enzymes | 111 | $0.178 \pm 0.019$ | $0.181 \pm 0.021$ | $0.097 \pm 0.014$ | $0.321 \pm 0.021$ | $\mathbf{0.340 \pm 0.022}$ |

the best method should be able to perform well across all sub-benchmarks. Table 6 shows the test performance of top performing methods on all sub-benchmarks at support set size 64 (the median of all considered support sizes) for both the classification and regression tasks. The results indicate that, in addition to achieving best overall performance, ADKF-IFT achieves the best performance on all sub-benchmarks for the regression task and on more than half of the sub-benchmarks for the classification task.

## F.4 META-TESTING COSTS

Figure 5 shows the meta-testing costs of all compared meta-learning methods in terms of wall-clock time[1] on a pre-defined set of FS-Mol classification tasks. These experiments are run on a single NVIDIA GeForce RTX 2080 Ti. It can be seen that ADKF-IFT is $\sim 2.5$x slower than CNP, ProtoNet, and DKT, but still much faster than GNN-MAML. We did not report the wall-clock time for PAR, because it is extremely memory intensive (PAR takes $> 10$x memory than ADKF-IFT does) and thus

---

[1]We acknowledge that wall-clock time may not be the best metric for measuring the costs, since some meta-learning methods could be parallelized, which will reduce the wall-clock time accordingly. An alternative metric is multiply–accumulate operation (MAC). However, it is difficult to obtain the accurate number of MACs due to the opaqueness of the GP modules used.

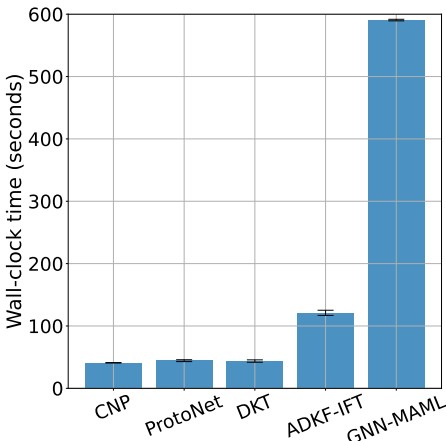

Figure 5: Wall-clock time consumed (with standard errors) when meta-testing on a pre-defined set of FS-Mol classification tasks using each of the compared meta-learning methods.

Table 7: Descriptions of four out-of-domain molecular design tasks.

| Molecular design task | Data source | #compounds | Target | Target source |
|---|---|---|---|---|
| Molecular docking (ESR2) | DOCKSTRING training set | 2,312 | binding score | AutoDock Vina |
| Antibiotic discovery (E. coli BW25113) | Antibiotic training set | 2,335 | relative growth | screening |
| Antiviral drug design (SARS-CoV-2) | COVID Moonshot | 1,926 | pIC50 Fluorescence | experimental lab |
| Material design (Organic Photovoltaic) | Harvard Clean Energy Project | 2,012 | power conversion efficiency | DFT simulation |

cannot be run on a GPU. We stress that this is not an important metric for this paper, as real-time adaptation is not required in drug discovery applications, but could be of interest if ADKF-IFT were to be deployed in other settings.

## G  DETAILS OF THE OUT-OF-DOMAIN MOLECULAR OPTIMIZATION EXPERIMENTS

In Table 7, we summarize the four molecular design tasks considered in Section 5.3. Note that the datasets for the molecular docking and material design tasks are subsampled from the much larger datasets provided in DOCKSTRING (García-Ortegón et al., 2021) and Harvard Clean Energy Project (Hachmann et al., 2011), respectively. The datasets for the antibiotic discovery and antiviral drug design tasks are taken from the antibiotic training set and the COVID Moonshot dataset provided in Stokes et al. (2020) and Consortium et al. (2022), respectively.

For the configuration of the GP, we use the Tanimoto kernel for fingerprint (with radius 2 and 2,048 bits based on count simulation) and Matérn52 kernel without ARD (with a log-normal prior over the lengthscale, centered at the median heuristic initialization) for all the other compared feature representations. We re-fit the base kernel parameters using all available data points at the beginning of each BO iteration.

## H  DISCUSSIONS OF METHODS FOR LEARNING DEEP KERNEL GPS

### H.1  THE OVERFITTING ISSUE IN DKL

Deep Kernel Learning (DKL) (Wilson et al., 2016b) is a single-task method for fitting a deep kernel to a dataset. DKL jointly fits both the feature extractor parameters $\phi$ and base kernel parameters $\theta$ by maximizing the GP marginal likelihood on a single dataset (i.e., DKL essentially fits a neural network with a GP "head" to a dataset).

It is well known that neural networks will easily overfit to small datasets (Sarle, 1995). This overfitting also happens in DKL, despite the fact that it fits the neural network parameters using a type-II

maximum likelihood approach: although early DKL papers suggested that the "model complexity" term (as measured by the log determinant of the kernel matrix) in the GP marginal likelihood objective would prevent this overfitting from happening (Wilson et al., 2016b), recent follow-up work showed that this is not the case (Ober et al., 2021) – a deep-kernel GP can simultaneously overfit to the training data and appear to have a low "model complexity".

## H.2 The Underfitting Issue in DKT

Deep Kernel Transfer (DKT) (Patacchiola et al., 2020) is a meta-learning method for fitting a deep kernel to a distribution of datasets. DKT jointly fits both the feature extractor parameters $\phi$ and base kernel parameters $\theta$ by maximizing the expected GP marginal likelihood over a distribution of datasets.

To mitigate the overfitting issue of DKL using meta-learning, DKT makes a very strong assumption that different tasks in the task distribution are drawn from an identical GP prior over functions. Explicitly, this means that the data generating process is assumed to have the same noise level, same amplitude, and same "characteristic lengthscale" for every task in the meta-dataset, which is a very restrictive assumption violated by most real-world problems. For example, different datasets in a meta-dataset may have

- highly varying noise levels, so modelling all tasks with the same amount of observation noise will not be realistic;
- different output ranges and units for regression: for example, one task might have data in the range 1-20 $\mu$M, while another might have 0-100% inhibition, meaning that a single signal variance (kernel amplitude) will not model the data well;
- different "characteristic lengthscales": for some tasks, structurally similar molecules have very strongly correlated output labels, while for other tasks it is much weaker (i.e., there is much more variation in the labels of very similar molecules), suggesting that the "characteristic lengthscale" will be different.

Inevitably, trying to fit such a misspecified model will result in a set of compromised base kernel parameters $\theta$ which fit all datasets okay on average but do not fit each individual dataset very well. This is the underfitting issue of DKT.

## H.3 The Advantage of ADKF-IFT

ADKF-IFT combines DKL and DKT in a way that can potentially inherit the strengths of both methods and the weaknesses of neither. By adapting the base kernel parameters $\theta$ specifically to each task, it prevents underfitting due to varying ranges, lengthscales, or noise levels between datasets. By meta-learning the feature extractor on many datasets, it prevents overfitting as observed by Ober et al. (2021). This advantage is both *theoretically principled* (by solving a bilevel optimization objective using the implicit function theorem) and *empirically observable* (we showed a statistically significant performance improvement of ADKF-IFT over DKT in Section 5.2).

# I Extended related work

## I.1 Multi-task Gaussian processes

ADKF-IFT can be considered a method to learn multi-task GPs. The dominant approach to this problem in prior works is to learn a shared kernel for all data points across all tasks, transmitting information by explicitly modelling the covariance between data from different tasks (Kennedy & O'Hagan, 2000; Forrester et al., 2007; Bonilla et al., 2007; Swersky et al., 2013; Poloczek et al., 2017). The main difference between methods in this family is the exact form of the kernel (Tighineanu et al., 2022), which is typically assumed to have a particular structure (e.g. Kronecker product or weighted sum of simpler kernels). ADKF-IFT cannot be naturally viewed in this way because the covariance between data points from separate tasks is always zero; information is instead transmitted between tasks via a shared set of kernel parameters. Therefore, we believe that ADKF-IFT is a significant departure from the dominant paradigm in GP transfer learning.

## I.2 IMPLICIT FUNCTION THEOREM IN MACHINE LEARNING

The implicit function theorem employed in our work has been used in many previous machine learning papers in various contexts, e.g., neural architecture search (Zhang et al., 2021), hyperparameter-tuning (Bengio, 2000; Luketina et al., 2016; Pedregosa, 2016; Lorraine et al., 2020; Clarke et al., 2022), and meta-learning (Rajeswaran et al., 2019; Lee et al., 2019; Chen et al., 2020).

## I.3 MODULAR META-LEARNING WITH SHRINKAGE (CHEN ET AL., 2020)

The method proposed by Chen et al. (2020) shares many similarities with ADKF-IFT: at a high level it also divides model parameters into meta-learned and adapted parameters[2], and optimizes the meta-learned parameters using the gradient of the validation loss after the adapted parameters have been adjusted to minimize the training loss using the implicit function theorem. The main differences between this work and ADKF-IFT are:

1. Model: Chen et al. (2020) consider a model where $\phi$ are the means and variances of a Gaussian prior over model parameters, whereas in ADKF-IFT $\phi$ is a subset of the parameters of an arbitrary deep kernel GP.

2. Hessian: Chen et al. (2020) consider the case where $\Theta$ is too large to form the exact Hessian for the implicit function theorem. They instead use a conjugate gradient approximation. Although some instantiations of ADKF-IFT could require this, in our highlighted version the Hessian can be computed exactly, which we view as a significant advantage.

3. Goal: The stated goal of Chen et al. (2020) is to decide which parameters should be meta-learned, while in ADKF-IFT this must be pre-specified, and we give guidance for doing so in a way that results in transferable meta-learned features.

## I.4 COMMENTS ON ADKL-GP (TOSSOU ET AL., 2019)

The preprint of Tossou et al. (2019) proposes an alternative adaptive deep kernel GP trained with meta-learning, where adaptation is performed by conditioning the feature extractor on an embedding of the entire support set rather than adjusting a subset of the kernel parameters as in ADKF-IFT. In general their empirical results were not very strong, and in our opinion the method is very prone to overfitting, which we explain below.

The training objective for ADKL-GP is equivalent to the objective for DKT with an added contrastive loss, weighted by a sensitive hyperparameter $\gamma$ (see Equation (13) of Tossou et al. (2019)). $\gamma$ can be interpreted as balancing the degree of regularization between two extremes:

1. If $\gamma = 0$, there is no regularization of the task encoding network, making significant overfitting to the meta-dataset possible. This is effectively equivalent to standard DKL (Wilson et al., 2016b).

2. As $\gamma \to \infty$, the regularization becomes infinitely strong, causing the task embeddings $\mathbf{z}_{\mathcal{T}}$ to collapse, and thereby preventing them from transmitting any information about specific datasets. With no information from $\mathbf{z}_{\mathcal{T}}$ in this case, the objective is essentially the same as that of DKT (Patacchiola et al., 2020).

For this method to be useful it would appear that $\gamma$ would need to be carefully tuned to balance between these extremes. Tossou et al. (2019) perform a grid search over all hyperparameters including $\gamma \in \{0, 0.01, 0.1\}$ but find no consistent trend besides $\gamma > 0$ being slightly helpful, although the differences in performance were small. This suggests that the method may be difficult to use in practice. ADKF-IFT however has no such tunable hyperparameters, which we view as a significant strength. Instead, the balance between DKL and DKT is controlled by selecting which parameters are adapted and meta-learned, which is much more interpretable and makes it easier to use in practice.

## J FUTURE WORK

Some directions for future work are as follows:

---

[2]They use the terms *meta parameters* and *task-specific parameters* instead.

1. using ARD in the base kernel so that feature selection for each individual task can be done by the GP model, with potential overfitting problems being reduced by assuming a sparse prior over lengthscales or by learning a low-dimensional manifold for them;

2. adapting the feature extractor to each task as well by allowing small deviations across tasks according to a meta-learned prior on the feature extractor parameters (e.g., as described in Chen et al. (2020));

3. adopting a more principled approximate inference strategy for few-shot GP classification (e.g., Pólya-Gamma data augmentation (Snell & Zemel, 2020) or Laplace approximation (Kim & Hospedales, 2021));

4. injecting domain expertise in drug discovery into the base kernel with hand-curated features and kernel combinations.

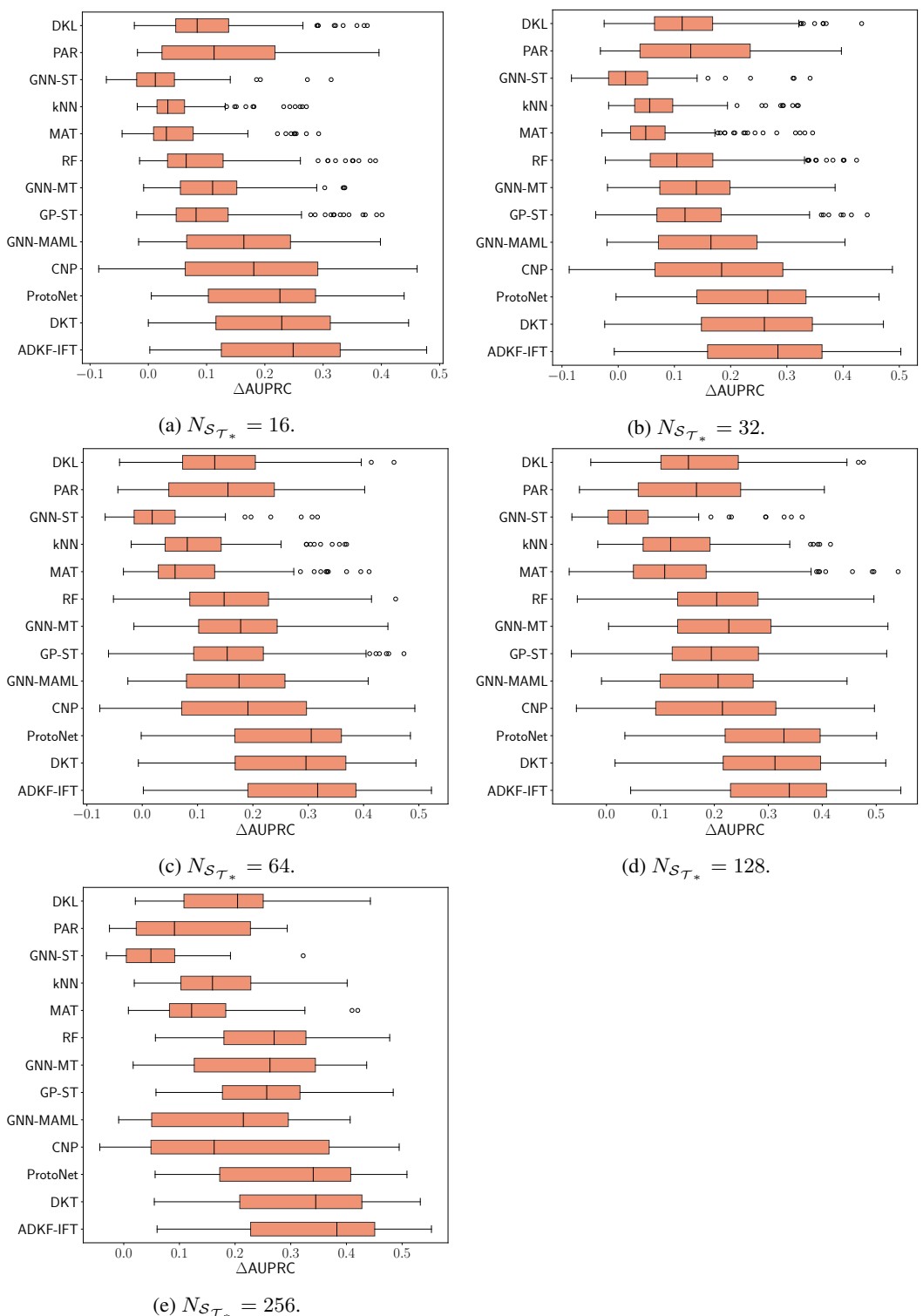

Figure 6: Box plots for the classification performance of all compared methods on 157 FS-Mol test tasks at different support set sizes.

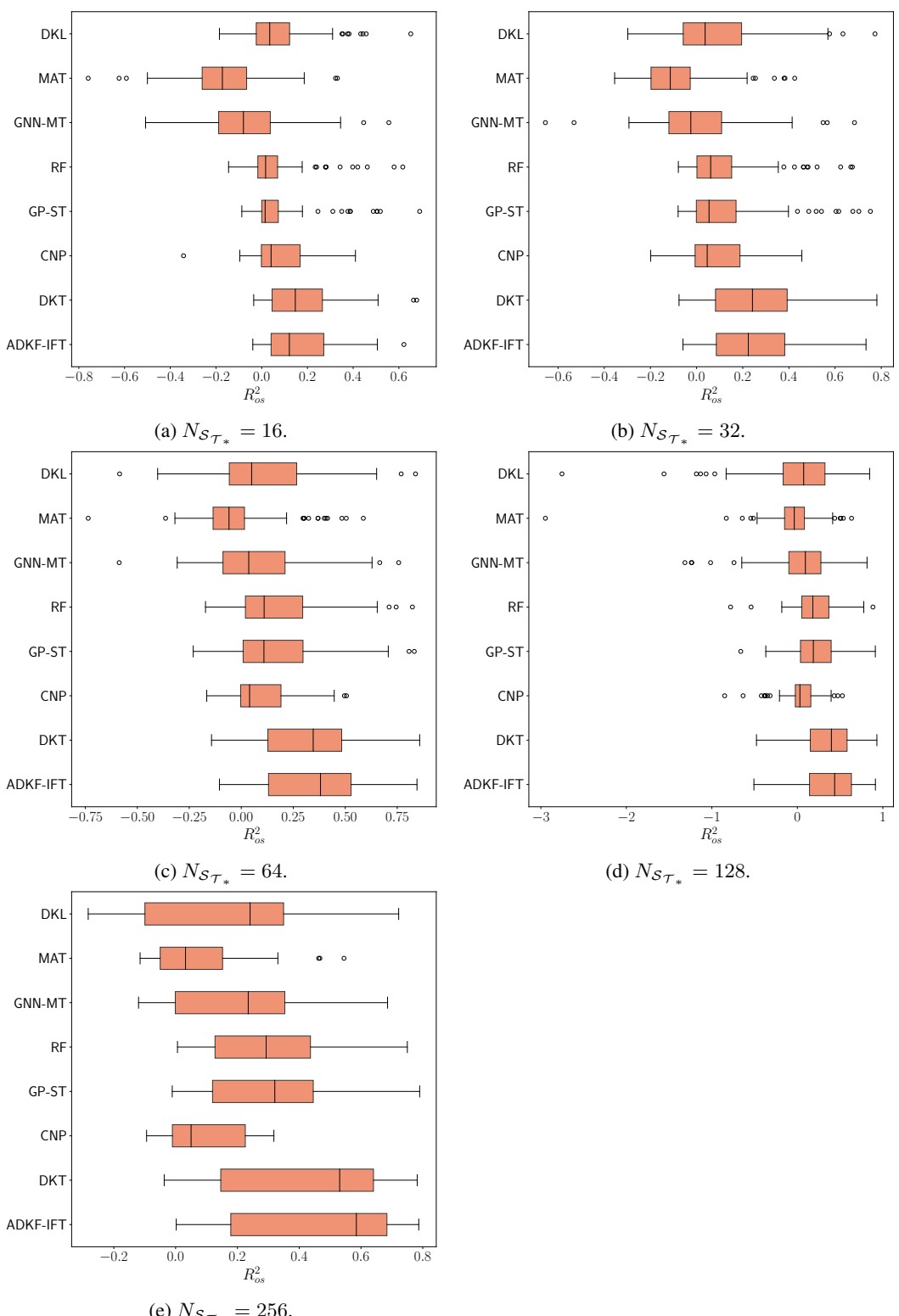

Figure 7: Box plots for the regression performance of all compared methods on 111 FS-Mol test tasks at different support set sizes.

