# OpenReview forum: "Meta-learning Adaptive Deep Kernel Gaussian Processes for Molecular Property Prediction"
_ICLR.cc/2023/Conference — ICLR 2023 poster_

### Official Review · Reviewer_fnkh · 2022-10-23

**Confidence:** 3
**Correctness:** 3
**Technical Novelty And Significance:** 2
**Empirical Novelty And Significance:** 3
**Recommendation:** 8

**Clarity, Quality, Novelty And Reproducibility:**

Except for the questions in 'Weaknesses', the paper is well-written in general. Its technical novelty efficiently combines existing methods and tools under a new meta-learning framework proposed by the authors. The empirical contribution of the paper seems quite significant by providing other few-shot learning baselines on FS-MOL for various support set sizes. It seems that implementation details are provided at a reasonable level of detail. Still, the reproducibility can be improved by disclosing the code.

**Strength And Weaknesses:**

### Strengths
- The paper proposed a meta-learning framework that is formulated via bilevel optimization and optimized using the implicit function theorem with general applicability.
- In its application to deep kernel models, the authors propose a deep kernel model with a meta-learned feature extractor and fine-tunable GP parameters. This not only circumvents the computational hassle of the proposed but also gives a quite interpretable hierarchical structure. While including existing methods, DKL and DKT as special cases, this specific model is a balanced interpolation of both under the proposed meta-learning framework.
- The proposed ADKF-IFT is extensively compared with existing methods on molecular property prediction few-shot learning.
Especially, in the experiments on FS-MOL benchmarks, it seems that much effort and resources have been put into the experiment, which provides also good baselines for others as the authors benefited from the baseline results in MoleculeNet experiments.
- In addition to its strengths in meta-learning, the authors also conduct Bayesian optimization experiments which supports the good quality of learned representation in the proposed meta-learning framework and the well-calibrated uncertainty of ADKF-IFT.




### Weaknesses
- Is a random initialization of $\psi_{adapt}$ enough to guarantee the below?
$$
IF(\psi_{adapt} | \psi_{meta}, S_T) = \frac{\partial L_{train} (\psi_{meta}, \psi_{adapt}, S_T)}{\partial \psi_{adapt}} = 0
$$
    - In Appendix B, it says that in ADKF-IFT, the fine-tuned parameter $\psi_{adapt}$ is initialized using the median heuristic.
    - It seems that such initialization does not guarantee $IF(\psi_{adapt} | \psi_{meta}, S_T) = 0$.\
Does the median heuristic generate a point $\psi_{adapt}$ such that $IF(\psi_{adapt} | \psi_{meta}, S_T) = 0$?\
Is there a mechanism that drives a point $\psi_{adapt}$ such that $IF(\psi_{adapt} | \psi_{meta}, S_T) \neq 0$ to a point such that $IF(\psi_{adapt} | \psi_{meta}, S_T) \neq 0$?
- How is it guaranteed that $\psi_{adapt}^*$ is a minimum?
    - From Thm.1, it seems that when the implicit function theorem is used, implicitly defining function is
$$
IF(\psi_{adapt} | \psi_{meta}, S_T) = \frac{\partial L_{train} (\psi_{meta}, \psi_{adapt}, S_T)}{\partial \psi_{adapt}} = 0
$$
This implies that for a given $\psi_{meta}$, $\psi_{adapt}$ satisfying the above equation is a maximum, a minimum, or a saddle point.
    - I was wondering how $\psi_{adapt}$ is guaranteed to be a minimum.\
Even if the initialization guarantees $IF(\psi_{adapt} | \psi_{meta}, S_T) = 0$ or the gradient descent eventually makes the point to satisfy $IF(\psi_{adapt} | \psi_{meta}, S_T) = 0$,
    - It seems unlikely that the minimization of upper validation loss is obtained with the maximization of lower train loss. \
This may imply that some care is necessary for the choice of two losses in this bilevel meta-learning framework.\
I agree that the specific instantiation of ADKF-IFT seems reasonable also in this respect.\
Still, I am curious about the authors' thoughts and opinions on this.
- Sensitivity to the initialization
    - Even if it is guaranteed that $\frac{\partial L_{train} (\psi_{meta}, \psi_{adapt}, S_T)}{\partial \psi_{adapt}} = 0$, it is imaginable that, in some case, $\psi_{adapt}$ is initialized to a point of a connected component of $\\{\psi_{adapt} | \frac{\partial L_{train} (\psi_{meta}, \psi_{adapt}, S_T)}{\partial \psi_{adapt}} = 0 \\}$ which consists of maximum points.
Considering this case, I am curious about the sensitivity of the method to the initialization or the choice of hyperparameters (some hyperparameters may exclude such bad cases).





### Minor points
- It seems that this somewhat seemingly relevant work is missing
    - Few-Shot Bayesian Optimization with Deep Kernel Surrogates; ICLR 2021
    - I am not asking to include this as an additional baseline.
    - The comparison with this seems to be able to further highlight the benefit of ADKF-IFT.
- Significance of the differences in the ablation study.
    - I can check visually the consistency of the superiority of ADKF-IFT in Figure 3. But in many cases, the confidence intervals (I guess they are from repeated experiments with some randomness, it is reasonable to call them confidence intervals) overlap. Maybe providing some simple statistical results may strengthen the argument of the ablation study.
- Duplicated names
    - In the paper, ADKF-IFT stands for two things, 1) meta-learning framework formulated by a bilevel optimization problem and trained using the implicit function theorem, 2) deep kernel GP model with meta-learned feature extractor and fine-tuned GP parameters using the proposed meta-learning framework
    - Saving ADKF-IFT to the proposed specific deep kernel methods, it may be less confusing to use another name for the proposed meta-learning framework reflecting its generality, for example, Meta-BIFT (meta-learning formulated via BIlevel optimization and optimized via Implicit Function Theorem)


**Summary Of The Paper:**

This paper proposes a new meta-learning framework formulated as a bilevel optimization problem. The implicit function theorem is utilized to efficiently compute the gradient of the bilevel optimization problem. It is shown that in the context of deep kernel learning the proposed framework encompasses DKL and DKT as extreme cases. In the wide range of models under this framework, the author proposes ADKF-IDF which meta-learns the parameters of the feature extractors and fine-tunes the parameters of the base kernel. This specific design choice avoids the computational hassle in the gradient computation and induces a natural interpretation of deep kernel learning in meta-learning. The ADKF-IDF is extensively compared to existing methods on few-shot learning tasks predicting molecular properties and on Bayesian optimization of molecular properties.

**Summary Of The Review:**

The paper proposes a new meta-learning framework and a specific instance of it that is shown to be effective in few-short learning and out-of-sample prediction demonstrated via Bayesian optimization experiment. With a good combination of many technical tools under the new meta-learning framework, the paper has good empirical contributions. As long as my concerns mentioned in 'Weakness' can be addressed, I am willing to increase my score and support the acceptance of the paper.

---

> ### Author Response · Authors · 2022-11-08
> **Response to the “weaknesses” section: clarification about zero gradient for applying IFT**
>
> Thanks for your detailed review of our paper! We will respond to your other comments shortly, but we wanted to promptly address the questions in the “weaknesses” section of your review. We suspect that there may be a misunderstanding of our method: we do not apply the Implicit Function Theorem (IFT) directly to the initialization of the parameters $\psi_{adapt}$. Instead, we run an optimizer like L-BFGS to converge to a point $\psi_{adapt}^*$ with $IF=0$ before applying IFT. Therefore, the initialization is only important insofar as it allows the optimizer to converge to a good local minimum. For GPs, initializing the lengthscale with the median distance has been shown to be an effective heuristic to achieve this (see e.g. [1, 2]). **We believe the reviewer may have thought that the IFT was being applied without optimization, which is not the case.** Below, we address your questions in detail individually:
>
> > Does the median heuristic generate a point $\psi_{adapt}$ such that $IF=0$?
>
> No, it will almost certainly have $IF\neq 0$. It is just used as a starting point for the L-BFGS optimizer. IFT is not applied to this initialization point. Instead, IFT is applied after the L-BFGS has converged.
>
> > Is there a mechanism that drives a point such that $IF\neq 0$ to a point such that $IF=0$?
>
> Yes, the L-BFGS optimizer (or in general any optimizer).
>
> > How is it guaranteed that $\psi_{adapt}$ is a minimum? I am curious about sensitivity of the method to the initialization or the choice of hyperparameters.
>
> While it is possible in principle that L-BFGS could converge to a saddle point or a local maximum, this should only happen if it is initialized in or very close to one of these locations, which is very unlikely to happen in practice (this has never been observed to happen with L-BFGS for GP base kernel parameter optimization). If it actually happened, saddle-free optimizers could be used (e.g. [3]). In addition, it has been shown that the median heuristic provides a good initialization for L-BFGS to converge to a good local minimum for GP base kernel parameters (see e.g. [1, 2]).
>
> > It is imaginable that, in some case, $\psi_{adapt}$  is initialized to a point of a connected component of {$\psi_{adapt}|IF=0$} which consists of maximum points.
>
> In our specific instantiation of ADKF-IFT, we can show that this particular bad scenario is **impossible**. This is because one of the parameters of the base kernel is the kernel amplitude (signal variance) $s$, and the derivative of the negative log marginal likelihood function (i.e. the train loss) with respect to $s$ has the form $\propto s^{-1} - ks^{-2}$ ($k>0$ is a constant), which has only a single critical point at $s=k$. The second derivative has the form $\propto -s^{-2} +2ks^{-3}$ which at $s=k$ is positive (since $k>0$), proving that this only critical point with respect to $s$ is a minimum. Therefore, any zero-gradient point will be a minimum along the $s$ direction (i.e., **our train loss does not have a maximum**). Reference for the above derivation can be found in https://proceedings.mlr.press/v161/ober21a.html, proof of proposition 1 in the supplementary material.
>
> > … some care is necessary for the choice of two losses in this bilevel meta-learning framework… I am curious about the authors' thoughts and opinions on this.
>
> In principle, one could specify a meta-learning problem in a bilevel optimization framework with arbitrary train (inner) loss and validation (outer) loss. However, the validation loss should be the model performance measurement that we care about, and this is what we optimize during meta-training (in our case, this is the negative log joint predictive posterior on the query set). The inner loss could again be anything, but we want to use it to adapt our model to the training data (in our case, this is the negative log marginal likelihood on the support set), and that is all we can do at meta-test time (we can no longer optimize the validation loss because we don’t have access to any labelled query data during meta-testing). Any reasonable optimizer (e.g. L-BFGS, Adam, etc) will be able to **minimize** any reasonable choice of the train loss (e.g., negative log marginal likelihood for GPs) and find a good $\psi_{adapt}^*$ with $IF=0$. Given this, the scenario that “the minimization of upper validation loss is obtained with the maximization of lower train loss” mentioned by the reviewer is impossible to happen.
>
> Please let us know if this response sufficiently addresses your main concerns for this paper.
>
> References:
>
> [1] Neil Houlsby, et al. "Collaborative gaussian processes for preference learning." Advances in neural information processing systems 25, 2012
>
> [2] Yingchao Xiao, et al. "Hyperparameter selection for Gaussian process one-class classification." IEEE transactions on neural networks and learning systems 26.9: 2182-2187, 2014
>
> [3] Martin Arjovsky, “Saddle-free Hessian-free Optimization”, arXiv:1506.00059, 2015

---

> > ### Comment · Reviewer_fnkh · 2022-11-09
> > **Thanks for the clarification!**
> >
> > The answers clarify the point I missed. With the application of IFT after first finding a critical point (mostly optimum) for $\phi_{adapt}$, the rest of my concerns become minor issues. Accordingly, I increase my score for the acceptance of this paper.

---

> ### Author Response · Authors · 2022-11-11
> **Response to "minor points" of reviewer fnkh**
>
> Thanks again for your comments on our paper! We are glad that our previous response has sufficiently addressed your main concerns for this paper. We will respond to your comments in the “minor points” section below:
>
> > It seems that this somewhat seemingly relevant work is missing.
>
> Thanks for pointing this out. We have cited this paper in the related work section in the revised paper. However, please note that the method described in this paper is very similar to DKT applied to the BO setting. As we already showed in our paper that DKT is a special case of our method in theory and empirically underperforms our method, we believe that this paper does not detract from our paper’s contribution.
>
> > Significance of the differences in the ablation study.
>
> Good suggestion. In the revised paper (Table 5 in Appendix F), we have added the p-values for the two-sided Wilcoxon signed-rank tests for statistical comparison between ADKF-IFT and the two ablation models (DKT+ and ADKF). In summary, ADKF-IFT significantly outperforms DKT+ and ADKF in most cases.
>
> > Duplicated names.
>
> Thanks for the suggestion. We are happy to use another name for the general framework in the camera-ready version. We decided not to change names during the review/discussion period, as that might cause confusion.
>
> We hope that we have sufficiently addressed all your concerns for this paper. Please let us know if you have any other questions or comments.

---

### Official Review · Reviewer_sNf8 · 2022-10-25

**Confidence:** 4
**Correctness:** 4
**Technical Novelty And Significance:** 3
**Empirical Novelty And Significance:** 3
**Recommendation:** 8

**Clarity, Quality, Novelty And Reproducibility:**

The paper is well-written and easy to understand. The work appears to be high-quality and well-executed, though no code has yet been shared.

Generalizing the DKL and DKT methods into a bilevel optimization problem in order to overcome the shortcomings of both is an original and important contribution.

**Strength And Weaknesses:**

Strengths:
- ADKF-IFT provides the means to effective training of Gaussian Processes that overcomes the shortcomings of the two methods it generalizes, DKL and DKT while achieving state-of-the-art predictive power.
- ADKF-IFT achieves state-of-the-art results on many molecular property prediction benchmarks, including few-shot tasks.
- ADKF-IFT appears to learn informative molecular features that perform well for entirely different tasks.

Weaknesses:
- The authors do not include any examples that demonstrate that ADKF-IFT is neither over- or underfit, nor do they demonstrate that the uncertainty estimates derived from the learned GPs are better than those of GPs learned in other ways.
- MoleculeNet is somewhat out-of-date - the authors should consider the superset of tasks proposed in the Therapeutics Data Commons (https://tdcommons.ai/) database.

**Summary Of The Paper:**

In order to fit molecular property prediction models that give good uncertainty estimates but also overcome issues associated with neural network methods that either overfit or underfit Gaussian Processes, the authors propose Adaptive Deep Kernel Fitting with Implicit Function Theorem (ADKF-IFT), a generalization of Deep Kernel Learning and Deep Kernel Transfer that meta-learns a subset of parameters (usually those associated with a feature extractor) across many tasks, and transfer learns the rest (usually those associated with the base kernels of Gaussian Processes). The authors propose a bilevel optimization method that avoids costly inner optimization gradient calculations by invoking Cauchy's Implicit Function Theorem. The authors demonstrate how ADKF-IFT outperforms a large variety of existing methods of molecular property prediction tasks in MoleculeNet and few-shot prediction tasks in FS-MOL. They also show how the feature representation learned by ADKF-IFT can be used with Gaussian Process surrogate models to provide optimal or near-optimal results in out-of-domain molecular optimization tasks like molecular docking and antiviral drug design. Using ablation studies, the authors also show that the bilevel optimization is what gives the reported performance gains.

**Summary Of The Review:**

This paper makes a significant contribution to Gaussian Process training methods by proposing a novel bilevel optimization method that achieves state-of-the-art molecular property prediction. Such models could have significant impact on the work of researchers applying ML to drug discovery, or other related topics.

---

> ### Author Response · Authors · 2022-11-14
> **Response to reviewer sNf8**
>
> Thanks for your comments on our paper! We will respond to your comments point by point below:
>
> > The authors do not include any examples that demonstrate that ADKF-IFT is neither over- or underfit. Nor do they demonstrate that the uncertainty estimates derived from the learned GPs are better than those of GPs learned in other ways.
>
> Overfitting and underfitting are difficult to precisely define and measure for probabilistic models. Concretely, our key claim is that our method can produce GPs with better uncertainty estimates than other GPs. We have demonstrated this in Section 5.3, where we performed Bayesian optimization (Figure 4) on four OOD tasks and also reported test log likelihood (Table 2) on these tasks. Both of these measures test the quality of the uncertainty estimates:
> - The test log likelihood rewards instances where the true value lies within the model’s outputted confidence interval, giving higher rewards to smaller intervals. Simultaneously, it penalizes instances where the true value lies outside the model’s outputted confidence interval, with larger penalties for greater distance from the confidence interval. This is a very direct measure of the quality of the uncertainty estimates.
> - Bayesian optimization (BO) provides a more practical evaluation of the model’s uncertainty estimates by testing how useful the estimates are for decision making. Roughly speaking, BO will avoid points with small uncertainty estimates, and will prioritize visiting points with large uncertainty estimates. Therefore, if the estimates are either too large or too small the performance will be poor.
>
> We compared ADKF-IFT against meta-learned GP (DKT), GP with a Tanimoto kernel (GP-ST), and GPs operating on top of features learned by other pretraining/meta-learning methods. The results showed that ADKF-IFT achieved best BO performance and best test log likelihood on most tasks, which we believe demonstrates that the uncertainty estimates derived from our learned GPs are better than those of other GPs.
>
> Given that other methods are known to be prone to overfitting or underfitting, we conjecture that our method’s success can be interpreted as achieving a successful balance between these two extremes. This is just a hypothesis, which is why we were careful to use the word _conjecture_ rather than _claim_ in the paper.
>
> > MoleculeNet is somewhat out-of-date - the authors should consider the superset of tasks proposed in the Therapeutics Data Commons database.
>
> We agree that MoleculeNet is not a very comprehensive benchmark. We used it to compare our method with the results of previous papers, not because we think it is a particularly good benchmark. We personally think that FS-Mol is the most convincing benchmark of our paper.
>
> Your suggestion of using TDC is a good idea, but adapting TDC into a meta-learning benchmark is non-trivial (e.g. we would need to define a reasonable split into meta-train and meta-test datasets, aggregate metrics, etc). We believe it would be good future work, but is not within the scope of our current paper.
>
> We hope that we have sufficiently addressed all your concerns for this paper. Please let us know if you have further questions or comments.

---

### Official Review · Reviewer_BY35 · 2022-11-02

**Confidence:** 3
**Correctness:** 3
**Technical Novelty And Significance:** 3
**Empirical Novelty And Significance:** Not applicable
**Recommendation:** 6

**Clarity, Quality, Novelty And Reproducibility:**

### Clarity
The writing is clear and easy to follow. The paper provides enough background to understand the paper.

### Novelty
The primary contributions of the paper are the interpolation between meta-learning & conventional deep kernel learning, and the use of implicit function theorem to solve the resulting bi-level optimization problem. These are both novel and interesting.

### Reproducibility
The paper does not provide all of the hyperparameter and architecture details to reproduce the results. However, the authors promise to make the code available upon publication.

**Strength And Weaknesses:**

### Strengths
* The paper presents a novel method for training deep kernel GPs that is competitive on small data domains.
* The proposed method can help balance over- and under-fitting.
* The paper demonstrates that the proposed method subsumes the existing DKT and DKL methods as special cases.

### Weaknesses
* It would be instructive to understand how the model performs on tasks with larger training datasets. I understand that is not the goal of the paper, but understanding the failure modes of the method (if there are any) can be useful.
* How sensitive is the method to the choice of \Psi_{adapt}, the parameters that are adapted to each task? Providing a discussion on this would be useful
* Provide more details about the hyperparameters & architectures for reproducing the results.

**Summary Of The Paper:**

The paper presents ADKF-IFT -- a new method to train deep kernel GPs using meta-learning. This method splits the parameters into two sets, where only one set is adapted to each task. This leads to a bi-level optimization problem, that the authors solve using the Implicit Function Theorem. The number of parameters which are adapted can help in balancing between over- and under-fitting, which helps the model perform well on tasks with small training datasets. The paper also shows that prior methods for training deep kernel GPs, DKT and DKL, are special cases of the proposed method.

Experimental results from chemistry show that the model generally performs competitively on small data domains, and also perform well on OOD tasks.

**Summary Of The Review:**

The paper presents a novel method for training deep kernel GPs which obtains strong results on small datasets. The proposed method is shown to be a generalization of existing methods. Overall, I find this work to be sufficiently novel and useful for many chemistry problems to justify acceptance.

---

> ### Author Response · Authors · 2022-11-11
> **Response to reviewer BY35**
>
> Thanks for your comments on our paper! We will respond to your comments point by point below:
>
> > It would be instructive to understand how the model performs on tasks with larger training datasets. I understand that is not the goal of the paper, but understanding the failure modes of the method (if there are any) can be useful.
>
> In the paper, we have shown that our model performs very well for tasks with training sets of size 256, which is quite large in the few-shot standard where people usually consider support sets of size 2-20. The use of GPs prevents us from applying our model on tasks with very large training sets, as training GPs requires inverting the kernel matrix for all training data. Furthermore, our method requires calculating the gradient of the kernel matrix with respect to its inputs, which makes scaling beyond O(1000) data points very difficult.
>
> As an example, we tested our model (with feature extractor meta-trained on FS-mol) on five molecular docking score regression tasks from the DockString benchmark [1]. These tasks have very large training sets of size ~221K. We adapt the base kernel parameters on 10K randomly sampled training data points, as this is the largest training set size that exact GPs can handle. The performance of ADKF-IFT was similar to that of a single-task GP trained on molecular fingerprints, and was much lower than the performance of a graph neural network. This is not surprising since neural networks are good at learning informative features from large datasets. We believe that the performance of ADKF-IFT could be improved by adapting the feature extractor (NN) parameters to each task as well. We are working on this idea as follow-up to this paper.
>
> > How sensitive is the method to the choice of $\Psi_{adapt}$, the parameters that are adapted to each task?
>
> We have not experimented with any other choice of $\Psi_{adapt}$. We expect that including all or part of the feature extractor (e.g., last few layers) in $\Psi_{adapt}$ would further improve the performance of our model on tasks with large training sets, but would overfit to tasks with small training sets. Approximation for the inverse Hessian in IFT is needed if there are too many (say >5K) parameters in $\Psi_{adapt}$.
>
> > Provide more details about the hyperparameters & architectures for reproducing the results.
>
> We have included all core details of hyperparameters in Appendix B in the revised paper. For MoleculeNet, we use GIN as the feature extractor (as mentioned in Section 5.1). For FS-Mol, the architecture details can be found in Appendix D. We will release the code and raw result data in the camera-ready version.
>
> Please let us know if this response has sufficiently addressed all your concerns and questions for this paper.
>
> Reference:
>
> [1] Miguel García-Ortegón, et al. "DOCKSTRING: easy molecular docking yields better benchmarks for ligand design." Journal of chemical information and modeling 62.15: 3486-3502, 2022

---

### Author Response · Authors · 2022-11-11
**General response to all reviewers**

We thank all reviewers for their valuable comments on our paper. We appreciate that **all three reviewers support the acceptance of our paper** and highlight the novelty, significance and relevance of our proposed method, our extensive experiments and promising empirical results, and our clear writing:

> **Reviewer BY35**: “Overall, I find this work to be sufficiently novel and useful for many chemistry problems to justify acceptance.”

> **Reviewer sNf8**: “This paper makes a significant contribution to Gaussian Process training methods by proposing a novel bilevel optimization method that achieves state-of-the-art molecular property prediction. Such models could have significant impact on the work of researchers applying ML to drug discovery, or other related topics.”

> **Reviewer fnkh**: “With a good combination of many technical tools under the new meta-learning framework, the paper has good empirical contributions.”


We will address each reviewer's concerns separately below their respective review. We have updated our manuscript to include some additional results and details requested by the reviewers:

1. As requested by reviewer BY35, we have included all core details of hyperparameters in Appendix B and details of architectures in Section 5.1 and Appendix D. We will release the code and raw result data in the camera-ready version.
2. As requested by reviewer fnkh, we have added statistical test results for ablation study in Table 5 (Appendix F), showing that ADKF-IFT significantly outperforms the two ablation models DKT+ and ADKF in most cases.
3. As suggested by reviewer fnkh, we have cited [1] in the related work section.

Reference:

[1] Martin Wistuba, Josif Grabocka. "Few-shot Bayesian optimization with deep kernel surrogates." ICLR 2021.

---

### Decision · Program_Chairs · 2023-01-20

**Decision:**

Accept: poster

**Justification For Why Not Higher Score:**

Still limitation to small dataset only; solving scalability issues and maintaining solid performance would have put the paper on the next score.


**Justification For Why Not Lower Score:**

Authors did a great job of summarizing highlights of the reviewers. All reviewers are in support for acceptance for the paper

- Reviewer BY35: “Overall, I find this work to be sufficiently novel and useful for many chemistry problems to justify acceptance.”
- Reviewer sNf8: “This paper makes a significant contribution to Gaussian Process training methods by proposing a novel bilevel optimization method that achieves state-of-the-art molecular property prediction. Such models could have significant impact on the work of researchers applying ML to drug discovery, or other related topics.”
- Reviewer fnkh: “With a good combination of many technical tools under the new meta-learning framework, the paper has good empirical contributions.”

Results are solid and provide interesting methodology to overcome issues in existing methodology.


**Metareview: Summary, Strengths And Weaknesses:**

Authors propose new meta-learning methods for deep kernel Gaussian processes called ADKF-IFT (Adaptive Deep Kernel Fitting with Implicit Function Theorem). Proposed method uses a meta-learning framework that is formulated via bi-level optimization and optimized using the implicit function theorem with general applicability. With ADKF-IFT, authors obtain SoTA results on small dataset molecular property prediction problems.

Strengths:
- Effective training methods for Gaussian Processes overcoming shortcomings of DKL, DKT
- Achieves state-of-the-art results on many molecular property prediction tasks including few-shot tasks
- Method learns informative molecular features that performs well on out-of-domain tasks
- Well written and easy to understand
- High quality and well executed
- Generalizing DKL and DKT into bilevel optimization overcoming issues of both is novel and important contribution
- Utilizing meta-learning (bi-level optimization) to interpolate between conventional deep kernel learning and meta-learning is an interesting and novel method.

Weakness
- MoleculeNet is out-of-date and a reviewer suggested new tasks (tasks proposed in the Therapeutics Data Commons database)
- No code for reproduction and details for reproduction is lacking (hparams, architecture details)
    - Author promised code upon publication
- Lack of evaluation of larger training size datasets.

AC encourages the authors to provide code and experiment as promised.


**Note From Pc:**

if the above contains the word "oral" or "spotlight" please see: "oral" presentation means -> notable-top-5% and "spotlight" means -> notable-top-25%. As stated in our emails, we are disassociating presentation type from AC recommendations